# Convergent Bregman Plug-and-Play Image Restoration for Poisson Inverse Problems

**Samuel Hurault**
Univ. Bordeaux, CNRS, INRIA, Bordeaux INP, IMB, UMR 5251
samuel.hurault@math.u-bordeaux.fr

**Ulugbek Kamilov**
Washington University in St. Louis
kamilov@wustl.edu

**Arthur Leclaire**
Univ. Bordeaux, CNRS, INRIA, Bordeaux INP, IMB, UMR 5251
LTCI, Télécom Paris, IP Paris
arthur.leclaire@telecom-paris.fr

**Nicolas Papadakis**
Univ. Bordeaux, CNRS, INRIA, Bordeaux INP, IMB, UMR 5251
nicolas.papadakis@math.u-bordeaux.fr

## Abstract

Plug-and-Play (PnP) methods are efficient iterative algorithms for solving ill-posed image inverse problems. PnP methods are obtained by using deep Gaussian denoisers instead of the proximal operator or the gradient-descent step within proximal algorithms. Current PnP schemes rely on data-fidelity terms that have either Lipschitz gradients or closed-form proximal operators, which is not applicable to Poisson inverse problems. Based on the observation that the Gaussian noise is not the adequate noise model in this setting, we propose to generalize PnP using the Bregman Proximal Gradient (BPG) method. BPG replaces the Euclidean distance with a Bregman divergence that can better capture the smoothness properties of the problem. We introduce the Bregman Score Denoiser specifically parametrized and trained for the new Bregman geometry and prove that it corresponds to the proximal operator of a nonconvex potential. We propose two PnP algorithms based on the Bregman Score Denoiser for solving Poisson inverse problems. Extending the convergence results of BPG in the nonconvex settings, we show that the proposed methods converge, targeting stationary points of an explicit global functional. Experimental evaluations conducted on various Poisson inverse problems validate the convergence results and showcase effective restoration performance.

## 1   Introduction

Ill-posed image inverse problems are classically formulated with a minimization problem of the form

$$\arg\min_{x\in\mathbb{R}^n} \lambda f(x) + g(x) \qquad (1)$$

where $f$ is a data-fidelity term, $g$ a regularization term, and $\lambda > 0$ a regularization parameter. The data-fidelity term is generally written as the negative log-likelihood $f(x) = -\log p(y|x)$ of the

37th Conference on Neural Information Processing Systems (NeurIPS 2023).

probabilistic observation model chosen to describe the physics of an acquisition $y \in \mathbb{R}^m$ from linear measurements $Ax$ of an image $x \in \mathbb{R}^n$. In applications such as Positron Emission Tomography (PET) or astronomical CCD cameras [Bertero et al., 2009], where images are obtained by counting particles (photons or electrons), it is common to use the Poisson noise model $y \sim \mathcal{P}(\alpha Ax)$ with parameter $\alpha > 0$. The corresponding negative log-likelihood corresponds to the Kullback-Leibler divergence

$$f(x) = \sum_{i=1}^{m} y_i \log\left(\frac{y_i}{\alpha(Ax)_i}\right) + \alpha(Ax)_i - y_i. \tag{2}$$

The minimization of (1) can be addressed with proximal splitting algorithms [Combettes and Pesquet, 2011]. Depending on the properties of the functions $f$ and $g$, they consist in alternatively evaluating the proximal operator and/or performing a gradient-descent step on $f$ and $g$.

Plug-and-Play (PnP) [Venkatakrishnan et al., 2013] and Regularization-by-Denoising (RED) [Romano et al., 2017] methods build on proximal splitting algorithms by replacing the proximal or gradient descent updates with off-the-shelf Gaussian denoising operators, typically deep neural networks trained to remove Gaussian noise. The intuition behind PnP/RED is that the proximal (resp. gradient-descent) mapping of $g$ writes as the maximum-a-posteriori (MAP) (resp. posterior mean) estimation from a Gaussian noise observation, under prior $p = \exp(-g)$. A remarkable property of these priors is that they are decoupled from the degradation model represented by $f$ in the sense that one learned prior $p$ can serve as a regularizer for many inverse problems. When using deep denoisers corresponding to exact gradient-step [Hurault et al., 2021, Cohen et al., 2021] or proximal [Hurault et al., 2022] maps, RED and PnP methods become real optimization algorithms with known convergence guarantees.

However, for Poisson noise the data-fidelity term (2) is neither smooth with a Lipschitz gradient, nor proximable for $A \neq \mathrm{Id}$ (meaning that its proximal operator cannot be computed in a closed form), which limits the use of standard splitting procedures. Bauschke et al. [2017] addressed this issue by introducing a Proximal Gradient Descent (PGD) algorithm in the Bregman divergence paradigm, called Bregman Proximal Gradient (BPG). The benefit of BPG is that the smoothness condition on $f$ for sufficient decrease of PGD is replaced by the "NoLip" condition "$Lh - f$ is convex" for a convex potential $h$. For instance, Bauschke et al. [2017] show that the data-fidelity term (2) satisfies the NoLip condition for the Burg's entropy $h(x) = -\sum_{i=1}^{n} \log(x_i)$.

Our primary goal is to exploit the BPG algorithm for minimizing (1) using PnP and RED priors. This requires an interpretation of the plug-in denoiser as a Bregman proximal operator.

$$\mathrm{Prox}_g^h(y) = \arg\min_x g(x) + D_h(x, y), \tag{3}$$

where $D_h(x, y)$ is the Bregman divergence associated with $h$. In Section 3, we show that by selecting a suitable noise model, other than the traditional Gaussian one, the MAP denoiser can be expressed as a Bregman proximal operator. Remarkably, the corresponding noise distribution belongs to an exponential family, which allows for a closed-form posterior mean (MMSE) denoiser generalizing the Tweedie's formula. Using this interpretation, we derive a RED prior tailored to the Bregman geometry. By presenting a prior compatible with the noise model, we highlight the limitation of the decoupling between prior and data-fidelity suggested in the existing PnP literature.

In order to safely use our MAP and MMSE denoisers in the BPG method, we introduce the Bregman Score denoiser, which generalizes the Gradient Step denoiser from [Hurault et al., 2021, Cohen et al., 2021]. Our denoiser provides an approximation of the log prior of the noisy distribution of images. Moreover, based on the characterization of the Bregman Proximal operator from Gribonval and Nikolova [2020], we show under mild conditions that our denoiser can be expressed as the Bregman proximal operator of an explicit nonconvex potential.

In Section 4, we use the Bregman Score Denoiser within RED and PnP methods and propose the B-RED and B-PnP algorithms. Elaborating on the results from Bolte et al. [2018] for the BPG algorithm in the nonconvex setting, we demonstrate that RED-BPG and PnP-BPG are guaranteed to converge towards stationary points of an explicit functional. We finally show in Section 5 the relevance of the proposed framework in the context of Poisson inverse problems.

## 2 Related Works

**Poisson Inverse Problems** A variety of methods have been proposed to solve Poisson image restoration from the Bayesian variational formulation (1) with Poisson data-fidelity term (2). Although

$f$ is convex, this is a challenging optimization problem as $f$ is non-smooth and does not have a closed-form proximal operator for $A \neq \mathrm{Id}$, thus precluding the direct application of splitting algorithms such as Proximal Gradient Descent or ADMM. Moreover, we wish to regularize (1) with an explicit denoising prior, which is generally nonconvex. PIDAL [Figueiredo and Bioucas-Dias, 2009, 2010] and related methods [Ng et al., 2010, Setzer et al., 2010] solve (2) using modified versions of the alternating direction method of multipliers (ADMM). As $f$ is proximable when $A = \mathrm{Id}$, the idea is to add a supplementary constraint in the minimization problem and to adopt an augmented Lagrangian framework. Figueiredo and Bioucas-Dias [2010] prove the convergence of their algorithm with convex regularization. However, no convergence is established for nonconvex regularization. Boulanger et al. [2018] adopt the primal-dual PDHG algorithm which also splits $A$ from the $f$ update. Once again, there is not convergence guarantee for the primal-dual algorithm with nonconvex regularization.

**Plug-and-Play (PnP)** PnP methods were successfully used for solving a variety of IR tasks by including deep denoisers into different optimization algorithms, including Gradient Descent [Romano et al., 2017], Half-Quadratic-Splitting [Zhang et al., 2017, 2021], ADMM [Ryu et al., 2019, Sun et al., 2021], and PGD [Kamilov et al., 2017, Terris et al., 2020]. Variations of PnP have been proposed to solve Poisson inverse problems. Rond et al. [2016] use PnP-ADMM and approximates, at each iteration, the non-tractable proximal operator of the data-fidelity term (2) with an inner minimization procedure. Sanghvi et al. [2022] propose a PnP version of the PIDAL algorithm which is also unrolled for image deconvolution. Theoretical convergence of PnP algorithms with deep denoisers has recently been addressed by a variety of studies [Ryu et al., 2019, Sun et al., 2021, Terris et al., 2020] (see also a review in Kamilov et al. [2023]). Most of these works require non-realistic or sub-optimal constraints on the deep denoiser, such as nonexpansiveness. More recently, convergence was addressed by making PnP genuine optimization algorithms again. This is done by building deep denoisers as exact gradient descent operators (*Gradient-Step denoiser*) [Cohen et al., 2021, Hurault et al., 2021] or exact proximal operators [Hurault et al., 2022, 2023]. These PnP algorithms thus minimize an explicit functional (2) with an explicit (nonconvex) deep regularization.

**Bregman optimization** Bauschke et al. [2017] replace the smoothness condition of PGD by the NoLip assumption (4) with a Bregman generalization of PGD called Bregman Proximal Gradient (BPG). In the nonconvex setting, Bolte et al. [2018] prove global convergence of the algorithm to a critical point of (1). The analysis however requires assumptions that are not verified by the Poisson data-fidelity term and Burg's entropy Bregman potential. Al-Shabili et al. [2022] considered unrolled Bregman PnP and RED algorithms, but without any theoretical convergence analysis. Moreover, the interaction between the data fidelity, the Bregman potential, and the denoiser was not explored.

## 3   Bregman denoising prior

The overall objective of this work is to efficiently solve ill-posed image restoration (IR) problems involving a data-fidelity term $f$ verifying the NoLip assumption for some convex potential $h$

$$\textbf{NoLip} \qquad \text{There is } L > 0 \text{ such that } Lh - f \text{ is convex on } int \operatorname{dom} h. \qquad (4)$$

PnP provides an elegant framework for solving ill-posed inverse problems with a denoising prior. However, the intuition and efficiency of PnP methods inherit from the fact that Gaussian noise is well suited for the Euclidean $L_2$ distance, the latter naturally arising in the MAP formulation of the Gaussian denoising problem. When the Euclidean distance is replaced by a more general Bregman divergence, the noise model needs to be adapted accordingly for the prior.

In Section 3.1, we first discuss the choice of the noise model associated to a Bregman divergence, leading to Bregman formulations of the MAP and MMSE estimators. Then we introduce in Section 3.2 the Bregman Score Denoiser that will be used to regularize the inverse problem (1).

**Notations** For convenience, we assume throughout our analysis that the convex potential $h : C_h \subseteq \mathbb{R}^n \to \mathbb{R} \cup \{+\infty\}$ is $\mathcal{C}^2$ and of Legendre type (definition in Appendix A.1). Its convex conjugate $h^*$ is then also $\mathcal{C}^2$ of Legendre type. $D_h(x, y)$ denotes its associated Bregman divergence

$$D_h : \mathbb{R}^n \times int \operatorname{dom} h \to [0, +\infty] : (x, y) \to \begin{cases} h(x) - h(y) - \langle \nabla h(y), x - y \rangle & \text{if } x \in \operatorname{dom}(h) \\ +\infty & \text{otherwise.} \end{cases} \qquad (5)$$

## 3.1 Bregman noise model

We consider the following observation noise model, referred to as *Bregman noise*[1],

$$\text{for } x, y \in \text{dom}(h) \times int\,\text{dom}(h) \quad p(y|x) := \exp\left(-\gamma D_h(x, y) + \rho(x)\right). \tag{6}$$

We assume that there is $\gamma > 0$ and a normalizing function $\rho : \text{dom}(h) \to \mathbb{R}$ such that the expression (6) defines a probability measure. For instance, for $h(x) = \frac{1}{2}||x||^2$, $\gamma = \frac{1}{\sigma^2}$ and $\rho = 0$, we retrieve the Gaussian noise model with variance $\sigma^2$. As shown in Section 5, for $h$ given by Burg's entropy, $p(y|x)$ corresponds to a multivariate Inverse Gamma ($\mathcal{IG}$) distribution.

Given a noisy observation $y \in int\,\text{dom}(h)$, *i.e.* a realization of a random variable $Y$ with conditional probability $p(y|x)$, we now consider two optimal estimators of $x$, the MAP and the posterior mean.

**Maximum-A-Posteriori (MAP) estimator** The MAP denoiser selects the mode of the a-posteriori probability distribution $p(x|y)$. Given the prior $p_X$, it writes

$$\hat{x}_{MAP}(y) = \arg\min_x -\log p(x|y) = \arg\min_x -\log p_X(x) - \log p(y|x) = \text{Prox}^h_{-\frac{1}{\gamma}(\rho + \log p_X)}(y). \tag{7}$$

Under the Bregman noise model (6), the MAP denoiser writes as the *Bregman proximal operator* (see relation (3)) of $-\frac{1}{\gamma}(\log p_X + \rho)$. This acknowledges for the fact that the introduced Bregman noise is the adequate noise model for generalizing PnP methods within the Bregman framework.

**Posterior mean (MMSE) estimator** The MMSE denoiser is the expected value of the posterior probability distribution and the optimal Bayes estimator for the $L_2$ score. Note that our Bregman noise conditional probability (6) belongs to the regular *exponential family of distributions*

$$p(y|x) = p_0(y) \exp\left(\langle x, T(y) \rangle - \psi(x)\right) \tag{8}$$

with $T(y) = \gamma \nabla h(y)$, $\psi(x) = \gamma h(x) - \rho(x)$ and $p_0(y) = \exp\left(\gamma h(y) - \gamma \langle \nabla h(y), y \rangle\right)$. It is shown in [Efron, 2011] (for $T = \text{Id}$ and generalized in [Kim and Ye, 2021] for $T \neq \text{Id}$) that the corresponding posterior mean estimator verifies a generalized Tweedie formula $\nabla T(y).\hat{x}_{MMSE}(y) = -\nabla \log p_0(y) + \nabla \log p_Y(y)$, which translates to (see Appendix B for details)

$$\hat{x}_{MMSE}(y) = \mathbb{E}[x|y] = y - \frac{1}{\gamma}(\nabla^2 h(y))^{-1} \cdot \nabla(-\log p_Y)(y). \tag{9}$$

Note that for the Gaussian noise model, we have $h(x) = \frac{1}{2}||x||^2$, $\gamma = 1/\sigma^2$ and (9) falls back to the more classical Tweedie formula of the Gaussian posterior mean denoiser $\hat{x} = y - \sigma^2 \nabla(-\log p_Y)(y)$.

Therefore, given an off-the-shelf "Bregman denoiser" $\mathcal{B}_\gamma$ specially devised to remove Bregman noise (6) of level $\gamma$, if the denoiser approximates the posterior mean $\mathcal{B}_\gamma(y) \approx \hat{x}_{MMSE}(y)$, then it provides an approximation of the score $-\nabla \log p_Y(y) \approx \gamma \nabla^2 h(y). (y - \mathcal{B}_\gamma(y))$.

## 3.2 Bregman Score Denoiser

Based on previous observations, we propose to define a denoiser following the form of the MMSE (9)

$$\mathcal{B}_\gamma(y) = y - (\nabla^2 h(y))^{-1}.\nabla g_\gamma(y), \tag{10}$$

with $g_\gamma : \mathbb{R}^n \to \mathbb{R}$ a nonconvex potential parametrized by a neural network. Such a denoiser corresponds to the Bregman generalization of the Gaussian Noise Gradient-Step denoiser proposed in [Hurault et al., 2021, Cohen et al., 2021].

When $\mathcal{B}_\gamma$ is trained as a denoiser for the associated Bregman noise (6) with $L_2$ loss, it approximates the optimal estimator for the $L_2$ score, precisely the MMSE (9). Comparing (10) and (9), we get $\nabla g_\gamma \approx -\frac{1}{\gamma}\nabla \log p_Y$, *i.e.* the score is properly approximated with an explicit conservative vector field. We refer to this denoiser as the *Bregman Score denoiser*.

**Is the Bregman Score Denoiser a Bregman proximal operator?** We showed in relation (7) that the optimal Bregman MAP denoiser $\hat{x}_{MAP}$ is a Bregman proximal operator. We want to generalize this

---

[1]The Bregman divergence being non-symmetric, the order of the variables $(x, y)$ in $D_h$ is important. Distributions of the form (6) with reverse order in $D_h$ have been characterized in [Banerjee et al., 2005] but this analysis does not apply here.

property to our Bregman denoiser (10). When trained with $L_2$ loss, the denoiser should approximate the MMSE rather than the MAP. For Gaussian noise, Gribonval [2011] re-conciliates the two views by showing that the Gaussian MMSE denoiser actually writes as an Euclidean proximal operator.

Similarly, extending the characterization from [Gribonval and Nikolova, 2020] of Bregman proximal operators, we now prove that, under some convexity conditions, the proposed Bregman Score Denoiser (10) explicitly writes as the Bregman proximal operator of a nonconvex potential.

**Proposition 1** (Proof in Appendix C). *Let $h$ be $\mathcal{C}^2$ and of Legendre type. Let $g_\gamma : \mathbb{R}^n \to \mathbb{R} \cup \{+\infty\}$ proper and differentiable and $\mathcal{B}_\gamma(y) : int\,\mathrm{dom}(h) \to \mathbb{R}^n$ defined in (10). Assume $\mathrm{Im}(\mathcal{B}_\gamma) \subset int\,\mathrm{dom}(h)$. With $\psi_\gamma : \mathbb{R}^n \to \mathbb{R} \cup \{+\infty\}$ defined by*

$$\psi_\gamma(y) = \begin{cases} -h(y) + \langle \nabla h(y), y \rangle - g_\gamma(y) & \text{if } y \in int\,\mathrm{dom}(h) \\ +\infty & \text{otherwise.} \end{cases} \tag{11}$$

*suppose that $\psi_\gamma \circ \nabla h^*$ is convex on $int\,\mathrm{dom}(h^*)$. Then for $\phi_\gamma : \mathbb{R}^n \to \mathbb{R} \cup \{+\infty\}$ defined by*

$$\phi_\gamma(x) := \begin{cases} g_\gamma(y) - D_h(x, y) \text{ for } y \in \mathcal{B}_\gamma^{-1}(x) & \text{if } x \in \mathrm{Im}(\mathcal{B}_\gamma) \\ +\infty & \text{otherwise} \end{cases} \tag{12}$$

*we have that for each $y \in int\,\mathrm{dom}(h)$*

$$\mathcal{B}_\gamma(y) \in \arg\min_{x \in \mathbb{R}^n} \{D_h(x, y) + \phi_\gamma(x)\} \tag{13}$$

**Remark 1.** *Note that the order in the Bregman divergence is important in order to fit the definition of the Bregman proximal operator (3). In this order, [Gribonval and Nikolova, 2020, Theorem 3] does not directly apply. We propose instead in Appendix C a new version of their main theorem.*

**Remark 2.** *As shown in Appendix C (equation (64)), $\mathcal{B}_\gamma(y)$ can be written as $\mathcal{B}_\gamma(y) = \nabla(\psi_\gamma \circ \nabla h^*) \circ \nabla h(y)$ so that the Jacobian $J_{\mathcal{B}_\gamma}(y) = \nabla^2 h(y).\nabla^2(\psi_\gamma \circ \nabla h^*)(\nabla h(y))$. The hypothesis of convexity of $\psi_\gamma \circ \nabla h^*$ on $int\,\mathrm{dom}(h^*)$ is thus equivalent to the fact that the Jacobian of $\mathcal{B}_\gamma$ is positive semi-definite on $int\,\mathrm{dom}(h)$.*

This proposition generalizes the result of [Hurault et al., 2022, Prop. 1] to any Bregman geometry, which proves that the Gradient-Step Gaussian denoiser writes as an Euclidean proximal operator when $\psi_\gamma \circ \nabla h^*(x) = \frac{1}{2}||x||^2 - g_\gamma(x)$ is convex. More generally, as exhibited for Poisson inverse problems in Section 5, such a convexity condition translates to a constraint on the deep potential $g_\gamma$.

To conclude, the Bregman Score Denoiser provides, via *exact* gradient or proximal mapping, two distinct explicit nonconvex priors $g_\gamma$ and $\phi_\gamma$ that can be used for subsequent PnP image restoration.

## 4 Plug-and-Play (PnP) image restoration with Bregman Score Denoiser

We now regularize inverse problems with the explicit prior provided by the Bregman Score Denoiser (10). Properties of the Bregman Proximal Gradient (BPG) algorithm are recalled in Section 4.1. We show the convergence of our two B-RED and B-PnP algorithms in Sections 4.2 and 4.3.

### 4.1 Bregman Proximal Gradient (BPG) algorithm

Let $F$ and $\mathcal{R}$ be two proper and lower semi-continuous functions with $F$ of class $\mathcal{C}^1$ on $int\,\mathrm{dom}(h)$. Bauschke et al. [2017] propose to minimize $\Psi = F + \mathcal{R}$ using the following BPG algorithm:

$$x^{k+1} \in \arg\min_{x \in \mathbb{R}^n} \{\mathcal{R}(x) + \langle x - x^k, \nabla F(x^k) \rangle + \frac{1}{\tau} D_h(x, x^k)\}. \tag{14}$$

Recalling the general expression of proximal operators defined in relation (3), when $\nabla h(x_k) - \tau \nabla F(x_k) \in \mathrm{dom}(h^*)$, the previous iteration can be written as (see Appendix D)

$$x^{k+1} \in \mathrm{Prox}_{\tau \mathcal{R}}^h \circ \nabla h^*(\nabla h - \tau \nabla F)(x_k). \tag{15}$$

With formulation (15), the BPG algorithm generalizes the Proximal Gradient Descent (PGD) algorithm in a different geometry defined by $h$.

**Convergence of BPG** If $F$ verifies the NoLip condition (4) for some $L_F > 0$ and if $\tau < \frac{1}{L_F}$, one can prove that the objective function $\Psi$ decreases along the iterates (15). Global convergence of the

iterates for nonconvex $F$ and $\mathcal{R}$ is also shown in [Bolte et al., 2018]. However, Bolte et al. [2018] take assumptions on $F$ and $h$ that are not satisfied in the context of Poisson inverse problems. For instance, $h$ is assumed strongly convex on the full domain $\mathbb{R}^n$ which is not satisfied by Burg's entropy. Additionally, $F$ is assumed to have Lipschitz-gradient on bounded subset of $\mathbb{R}^n$, which is not true for $F$ the Poisson data-fidelity term (2). Following the same structure of their proof, we extend in Appendix D.1 (Proposition 3 and Theorem 3) the convergence theory from [Bolte et al., 2018] with the more general Assumption 1 below, which is verified for Poisson inverse problems (Appendix E.4).

**Application to the IR problem** In what follows, we consider two variants of BPG for minimizing (1) with gradient updates on the data-fidelity term $f$. These algorithms respectively correspond to Bregman generalizations of the RED Gradient-Descent (RED-GD) [Romano et al., 2017] and the Plug-and-Play PGD algorithms. For the rest of this section, we consider the following assumptions

**Assumption 1.**

    *(i)* $h : C_h \to \mathbb{R} \cup \{+\infty\}$ *is of class $\mathcal{C}^2$ and of Legendre-type.*

    *(ii)* $f : \mathbb{R}^n \to \mathbb{R} \cup \{+\infty\}$ *is proper, lower-bounded, coercive, of class $\mathcal{C}^1$ on $int \operatorname{dom}(h)$, with $\operatorname{dom}(h) \subset \operatorname{dom}(f)$, and is subanalytic.*

    *(iii)* **NoLip :** $L_f h - f$ *is convex on $int \operatorname{dom}(h)$.*

    *(iv)* $h$ *is assumed strongly convex on any bounded convex subset of its domain and for all $\alpha > 0$, $\nabla h$ and $\nabla f$ are Lipschitz continuous on $\{x \in \operatorname{dom}(h), \Psi(x) \leq \alpha\}$.*

    *(v)* $g_\gamma$ *given by the Bregman Score Denoiser* (10) *and its associated $\phi_\gamma$ obtained from Proposition 1 are lower-bounded and subanalytic.*

Even though the Poisson data-fidelity term (2) is convex, our convergence results also hold for more general nonconvex data-fidelity terms. Assumption (iv) generalizes [Bolte et al., 2018, Assumption D] and allows to prove global convergence of the iterates. The subanalytic assumption, defined in Appendix A.3, is a sufficient condition for the Kurdyka-Lojasiewicz (KL) property to be verified [Bolte et al., 2007]. The latter, also defined in Appendix A.3, can be interpreted as the fact that, up to a reparameterization, the function is sharp. The KL property is widely used in nonconvex optimization [Attouch et al., 2013, Ochs et al., 2014, Bolte et al., 2018]. As detailed in Appendix A.3, the subanalytic assumption of $f$, $g_\gamma$ and $\phi_\gamma$ allows for the Kurdyka-Lojasiewicz (KL) property to be verified by the objective functions $f + g_\gamma$ and $f + \phi_\gamma$.

### 4.2 Bregman Regularization-by-Denoising (B-RED)

We first generalize the RED Gradient-Descent (RED-GD) algorithm [Romano et al., 2017] in the Bregman framework. Classically, RED-GD is a simple gradient-descent algorithm applied to the functional $\lambda f + g_\gamma$ where the gradient $\nabla g_\gamma$ is assumed to be implicitly given by an image denoiser $\mathcal{B}_\gamma$ (parametrized by $\gamma$) via $\nabla g_\gamma = \operatorname{Id} - \mathcal{B}_\gamma$. Instead, our Bregman Score Denoiser (10) provides an explicit regularizing potential $g_\gamma$ whose gradient approximates the score via the Tweedie formula (9). We propose to minimize $F_{\lambda, \gamma} = \lambda f + g_\gamma$ on $\operatorname{dom}(h)$ using the Bregman Gradient Descent algorithm

$$x_{k+1} = \nabla h^*(\nabla h - \tau \nabla F_{\lambda, \gamma})(x_k) \tag{16}$$

which writes in a more general version as the BPG algorithm (14) with $\mathcal{R} = 0$

$$x_{k+1} = \underset{x \in \mathbb{R}^n}{\arg\min} \{\langle x - x_k, \lambda \nabla f(x_k) + \nabla g_\gamma(x_k)\rangle + \frac{1}{\tau} D_h(x, x_k)\}. \tag{17}$$

As detailed in Appendix E.4, in the context of Poisson inverse problems, for $h$ being Burg's entropy, $F_{\lambda, \gamma} = \lambda f + g_\gamma$ verifies the NoLip condition only on *bounded* convex subsets of $dom(h)$. Thus we select $C$ a non-empty closed bounded convex subset of $\overline{\operatorname{dom}(h)}$. For the algorithm (17) to be well-posed and to verify a sufficient decrease of $(F_{\lambda, \gamma}(x^k))$, the iterates need to verify $x_k \in C$. We propose to modify (17) as the Bregman version of Projected Gradient Descent, which corresponds to the BPG algorithm (14) with $\mathcal{R} = i_C$, the characteristic function of the set $C$:

**(B-RED)**    $x^{k+1} \in T_\tau(x_k) = \underset{x \in \mathbb{R}^n}{\arg\min} \{i_C(x) + \langle x - x^k, \nabla F_{\lambda, \gamma}(x^k)\rangle + \frac{1}{\tau} D_h(x, x^k)\}.$    (18)

For general convergence of B-RED, we need the following assumptions

**Assumption 2.**

    *(i) $\mathcal{R} = i_C$, with $C$ a non-empty closed, bounded, convex and semi-algebraic subset of $\overline{\mathrm{dom}(h)}$ such that $C \cap int\, \mathrm{dom}(h) \neq \emptyset$.*

    *(ii) $g_\gamma$ has Lipschitz continuous gradient and there is $L_\gamma > 0$ such that $L_\gamma h - g_\gamma$ is convex on $C \cap int\, \mathrm{dom}(h)$.*

In [Bolte et al., 2018, Bauschke et al., 2017], the NoLip constant $L$ needs to be known to set the stepsize of the BPG algorithm as $\tau L < 1$. In practice, the NoLip constant which depends on $f$, $g_\gamma$ and $C$ is either unknown or over-estimated. In order to avoid small stepsize, we adapt the **backtracking** strategy of [Beck, 2017, Chapter 10] to automatically adjust the stepsize while keeping convergence guarantees. Given $\gamma \in (0,1)$, $\eta \in [0,1)$ and an initial stepsize $\tau_0 > 0$, the following backtracking update rule on $\tau$ is applied at each iteration $k$:

$$\text{while} \quad F_{\lambda,\gamma}(x_k) - F_{\lambda,\gamma}(T_\tau(x_k)) < \frac{\gamma}{\tau} D_h(T_\tau(x_k), x_k), \quad \tau \longleftarrow \eta\tau. \tag{19}$$

Using the general nonconvex convergence analysis of BPG realized in Appendix D.1, we can show sufficient decrease of the objective and convergence of the iterates of B-RED.

**Theorem 1** (Proof in Appendix D.2). *Under Assumption 1 and Assumption 2, the iterates $(x_k)$ given by the B-RED algorithm* (18) *with the backtracking procedure* (19) *decrease $F_{\lambda,\gamma}$ and converge to a critical point of $\Psi = i_C + F_{\lambda,\gamma}$ with rate $\min_{0 \leq k \leq K} D_h(x^{k+1}, x^k) = O(1/K)$.*

### 4.3 Bregman Plug-and-Play (B-PnP)

We now consider the equivalent of PnP Proximal Gradient Descent algorithm in the Bregman framework. Given a denoiser $\mathcal{B}_\gamma$ with $\mathrm{Im}(\mathcal{B}_\gamma) \subset \mathrm{dom}(h)$ and $\lambda > 0$ such that $\mathrm{Im}(\nabla h - \lambda \nabla f) \subseteq \mathrm{dom}(\nabla h^*)$, it writes

$$\textbf{(B-PnP)} \quad x^{k+1} = \mathcal{B}_\gamma \circ \nabla h^*(\nabla h - \lambda \nabla f)(x_k). \tag{20}$$

We use again as $\mathcal{B}_\gamma$ the Bregman Score Denoiser (10). With $\psi_\gamma$ defined from $g_\gamma$ as in (11) and assuming that $\psi_\gamma \circ \nabla h^*$ is convex on $int\, \mathrm{dom}(h^*)$, Proposition 1 states that the Bregman Score denoiser $\mathcal{B}_\gamma$ is the Bregman proximal operator of some nonconvex potential $\phi_\gamma$ verifying (12). The algorithm B-PnP (20) then becomes $x^{k+1} \in \mathrm{Prox}_{\phi_\gamma}^h \circ \nabla h^*(\nabla h - \lambda \nabla f)(x_k)$, which writes as a Bregman Proximal Gradient algorithm, with stepsize $\tau = 1$,

$$x^{k+1} \in \underset{x \in \mathbb{R}^n}{\arg\min}\{\phi_\gamma(x) + \langle x - x^k, \lambda \nabla f(x^k)\rangle + D_h(x, x^k)\}. \tag{21}$$

With Proposition 1, we have $\mathcal{B}_\gamma(y) \in \mathrm{Prox}_{\phi_\gamma}^h(y)$ *i.e.* a proximal step on $\phi_\gamma$ *with stepsize* 1. We are thus forced to keep a fixed stepsize $\tau = 1$ in the BPG algorithm (21) and no backtracking is possible. Using Appendix D.1, we can show that B-PnP converges towards a stationary point of $\lambda f + \phi_\gamma$.

**Theorem 2** (Proof in Appendix D.3). *Assume Assumption 1 and $\psi_\gamma \circ \nabla h^*$ strictly convex on $int\, \mathrm{dom}(h^*)$. Then for $\mathrm{Im}(\nabla h - \lambda \nabla f) \subseteq \mathrm{dom}(\nabla h^*)$, $\mathrm{Im}(\mathcal{B}_\gamma) \subseteq \mathrm{dom}(h)$ and $\lambda L_f < 1$ (with $L_f$ specified in Assumption 1), the iterates $x_k$ given by the B-PnP algorithm* (20) *decrease $\lambda f + \phi_\gamma$ and converge to a critical point of $\lambda f + \phi_\gamma$ with rate $\min_{0 \leq k \leq K} D_h(x^{k+1}, x^k) = O(1/K)$.*

**Remark 3.** *The condition $\mathrm{Im}(\mathcal{B}_\gamma) \subseteq \mathrm{dom}(h)$ and the required convexity of $\psi_\gamma \circ \nabla h^*$ come from Proposition 1 while the condition $\mathrm{Im}(\nabla h - \lambda \nabla f) \subseteq \mathrm{dom}(\nabla h^*)$ allows the algorithm B-PnP* (20) *to be well-posed. These assumptions will be discussed with more details in the context of Poisson image restoration in Section 5.*

## 5 Application to Poisson inverse problems

We consider ill-posed inverse problems involving the Poisson data-fidelity term $f$ introduced in (2). The Euclidean geometry (*i.e.* $h(x) = \frac{1}{2}||x||^2$) does not suit for such $f$, as it does not have a Lipschitz gradient. In [Bauschke et al., 2017, Lemma 7], it is shown that an adequate Bregman potential $h$ (in the sense that there exists $L_f$ such that $L_f h - f$ is convex) for (2) is the Burg's entropy

$$h(x) = -\sum_{i=1}^{n} \log(x_i), \tag{22}$$

for which $\text{dom}(h) = \mathbb{R}_{++}^n$ and $L_f h - f$ is convex on $int\,\text{dom}(h) = \mathbb{R}_{++}^n$ for $L_f \geq ||y||_1$. For further computation, note that the Burg's entropy (22) satisfies $\nabla h(x) = \nabla h^*(x) = -\frac{1}{x}$ and $\nabla^2 h(x) = \frac{1}{x^2}$. The Bregman score denoiser associated to the Burg's entropy is presented in Section 5.1. The corresponding Bregman RED and PnP algorithms are applied to Poisson Image deblurring in Section 5.2.

## 5.1  Bregman Score Denoiser with Burg's entropy

We now specify the study of Section 3 to the case of the Burg's entropy (22). In this case, the Bregman noise model (6) writes (see Appendix E.1 for detailed calculus)

$$p(y|x) = \exp(\rho(x) + n\gamma) \prod_{i=1}^n \left(\frac{x_i}{y_i}\right)^\gamma \exp\left(-\gamma \frac{x_i}{y_i}\right). \tag{23}$$

For $\gamma > 1$, this is a product of *Inverse Gamma* ($\mathcal{IG}(\alpha, \beta)$) distributions with parameter $\beta_i = \gamma x_i$ and $\alpha_i = \gamma - 1$. This noise model has mean (for $\gamma > 2$) $\frac{\gamma}{\gamma-2}x$ and variance (for $\gamma > 3$) $\frac{\gamma^2}{(\gamma-2)^2(\gamma-3)}x^2$. In particular, for large $\gamma$, the noise becomes centered on $x$ with signal-dependent variance $x^2/\gamma$. Furthermore, using Burg's entropy (22), the optimal posterior mean (9) and the Bregman Score Denoiser (10) respectively write, for $y \in \mathbb{R}_{++}^n$,

$$\hat{x}_{MMSE}(y) = y - \frac{1}{\gamma}y^2 \nabla(-\log p_Y)(y) \tag{24}$$

$$\mathcal{B}_\gamma(y) = y - y^2 \nabla g_\gamma(y). \tag{25}$$

**Denoising in practice** As Hurault et al. [2021], we chose to parametrize the deep potential $g_\gamma$ as

$$g_\gamma(y) = \frac{1}{2}||x - N_\gamma(x)||^2, \tag{26}$$

where $N_\sigma$ is the deep convolutional neural network architecture DRUNet [Zhang et al., 2021] that contains Softplus activations and takes the noise level $\gamma$ as input. $\nabla g_\gamma$ is computed with automatic differentiation. We train $\mathcal{B}_\gamma$ to denoise images corrupted with random Inverse Gamma noise of level $\gamma$, sampled from clean images via $p(y|x) = \prod_{i=1}^n \mathcal{IG}(\alpha_i, \beta_i)(x_i)$. To sample $y_i \sim \mathcal{IG}(\alpha_i, \beta_i)$, we sample $z_i \sim \mathcal{G}(\alpha_i, \beta_i)$ and take $y_i = 1/z_i$. Denoting as $p$ the distribution of a database of clean images, training is performed with the $L^2$ loss

$$\mathcal{L}(\gamma) = \mathbb{E}_{x \sim p, y \sim \mathcal{IG}_\gamma(x)} \left[||\mathcal{B}_\gamma(y) - x||^2\right]. \tag{27}$$

**Denoising performance** We evaluate the performance of the proposed Bregman Score DRUNet (B-DRUNet) denoiser (26). It is trained with the loss (27), with $1/\gamma$ uniformly sampled in $(0, 0.1)$. More details on the architecture and the training can be found in Appendix E.2. We compare the performance of B-DRUNet (26) with the same network DRUNet directly trained to denoise inverse Gamma noise with $L_2$ loss. Qualitative and quantitative results presented in Figure 1 and Table 1 show that the Bregman Score Denoiser (B-DRUNet), although constrained to be written as (25) with a conservative vector field $\nabla g_\gamma$, performs on par with the unconstrained denoiser (DRUNet).

| $\gamma$ | 10 | 25 | 50 | 100 | 200 |
|---|---|---|---|---|---|
| DRUNet | 28.42 | 30.91 | 32.80 | 34.76 | 36.79 |
| B-DRUNet | 28.38 | 30.88 | 32.76 | 34.74 | 36.71 |

Table 1: Average denoising PSNR performance of Inverse Gamma noise denoisers B-DRUNet and DRUNet on $256 \times 256$ center-cropped images from the CBSD68 dataset, for various noise levels $\gamma$.

**Bregman Proximal Operator** Considering the Burg's entropy (22) in Proposition 1 we get that, if $\eta_\gamma : x \to \psi_\gamma \circ \nabla h^*(x) = \psi_\gamma(-\frac{1}{x})$ is convex on $\text{dom}(h^*) = \mathbb{R}_{--}$, the Bregman Score Denoiser (25) satisfies $\mathcal{B}_\gamma(y) = \text{Prox}_{\phi_\gamma}^h$, for the nonconvex potential $\psi_\gamma(y) = \sum_{i=1}^n \log(y_i) - g_\gamma(y) - 1$. The convexity of $\psi_\gamma \circ \nabla h^*$ can be verified using the following characterization, which translates to a condition on the deep potential $g_\gamma$ (see Appendix E.2 for details)

$$\forall x \in \text{dom}(h^*), \ \forall d \in \mathbb{R}^n, \ \langle \nabla^2 \eta_\gamma(x)d, d\rangle \geq 0 \tag{28}$$

$$\Leftrightarrow \forall y \in \mathbb{R}_{++}^n, \ \forall d \in \mathbb{R}^n, \ \langle y^4 \nabla^2 g_\gamma(y)d, d\rangle \leq \sum_{i=1}^n \left(y^2(1 - 2y\nabla g_\gamma(y))\right)_i d_i^2 \tag{29}$$

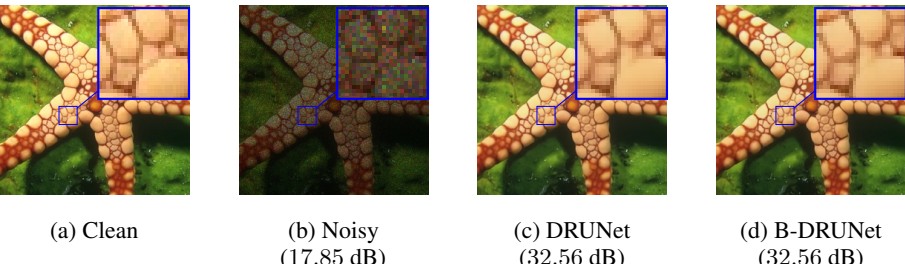

| (a) Clean | (b) Noisy | (c) DRUNet | (d) B-DRUNet |
|---|---|---|---|
| | (17.85 dB) | (32.56 dB) | (32.56 dB) |

Figure 1: Denoising of a $256 \times 256$ image corrupted with Inverse Gamma noise of level $\gamma = 25$.

It is difficult to constrain $g_\gamma$ to satisfy the above condition. In Appendix E.2, we proceed to a detailed experimental validation of this assumption. We empirically observe on images that the trained deep potential $g_\gamma$ satisfies the above inequality. This suggests that the convexity condition for Proposition 1 is true at least locally, on the image manifold. Indeed, we prove in Appendix E.3 that the convexity condition holds for the theoretical MMSE denoiser (24). As explained below, our denoiser is trained by minimizing the $L^2$ cost for which the optimal Bayes estimator is the MMSE. The MMSE is then a theoretical denoiser that our denoiser tries to approximate. This explains why our denoiser, after training, naturally satisfies this condition without necessitating supplementary constraints.

## 5.2 Bregman Plug-and-Play for Poisson Image Deblurring

We now derive the explicit B-RED and B-PnP algorithms in the context of Poisson image restoration. Choosing $C = [0, R]^n$ for some $R > 0$, the B-RED and B-PnP algorithms (18) and (20) write

$$\textbf{(B-RED)} \quad x_i^{k+1} = \arg \min \{ x \nabla F_{\lambda, \gamma}(x^k)_i + \frac{1}{\tau} \left( \frac{x}{x_i^k} - \log \frac{x}{x_i^k} \right) : x \in [0, R] \} \quad 1 \le i \le n \quad (30)$$

$$= \begin{cases} \frac{x_i^k}{1 + \tau x_i^k \nabla F_{\lambda, \gamma}(x^k)_i} & \text{if } 0 \le \frac{x_i^k}{1 + \tau x_i^k \nabla F_{\lambda, \gamma}(x^k)_i} \le R \\ R & \text{else} \end{cases} \quad 1 \le i \le n \quad (31)$$

$$\textbf{(B-PnP)} \quad x^{k+1} = \mathcal{B}_\gamma \left( \frac{x^k}{1 + \tau x^k \nabla f(x^k)} \right). \quad (32)$$

Verification of the assumptions of Theorems 1 and 2 for the convergence of both algorithms is discussed in Appendix E.4.

**Poisson deblurring** Equipped with the Bregman Score Denoiser, we now investigate the practical performance of the plug-and-play B-RED and B-PnP algorithms for image deblurring with Poisson noise. In this context, the degradation operator $A$ is a convolution with a blur kernel. We verify the efficiency of both algorithms over a variety of blur kernels (real-world camera shake, uniform and Gaussian). The hyper-parameters $\gamma$, $\lambda$ are optimized for each algorithm and for each noise level $\alpha$ by grid search and are given in Appendix E.5. In practice, we first initialize the algorithm with 100 steps with large $\tau$ and $\gamma$ so as to quickly initialize the algorithm closer to the right stationary point.

Note that the constraint $\lambda L_f < 1$ for convergence of the B-PnP algorithm (Theorem 1) may not be respected. The global NoLip constant $L_f$ can indeed be locally very lose. As explained in the Appendix (32), we can adopt a backtracking-like strategy on the regularization parameter $\lambda$ to ensure convergence. Nevertheless, with the proposed default value of $\lambda$, this backtracking algorithm was never activated over the variety of blur kernels and noise levels experimented.

We show in Figures 2 and 3 that both B-PnP and B-RED algorithms provide good visual reconstruction. Moreover, we observe that, in practice, both algorithms satisfy the sufficient decrease property of the objective function as well as the convergence of the iterates. Additional quantitative performance and comparisons with other Poisson deblurring methods are given in Appendix E.4.

## 6 Conclusion

In this paper, we derive a complete extension of the plug-and-play framework in the general Bregman paradigm for non-smooth image inverse problems. Given a convex potential $h$ adapted to the geometry

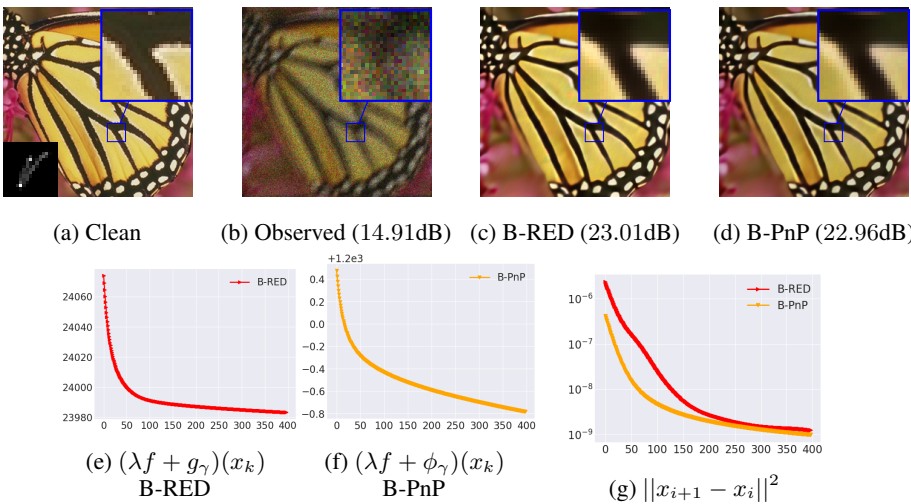

(a) Clean     (b) Observed (14.91dB)     (c) B-RED (23.01dB)     (d) B-PnP (22.96dB)

(e) $(\lambda f + g_\gamma)(x_k)$
B-RED

(f) $(\lambda f + \phi_\gamma)(x_k)$
B-PnP

(g) $||x_{i+1} - x_i||^2$

Figure 2: Deblurring from the indicated motion kernel and Poisson noise with $\alpha = 40$.

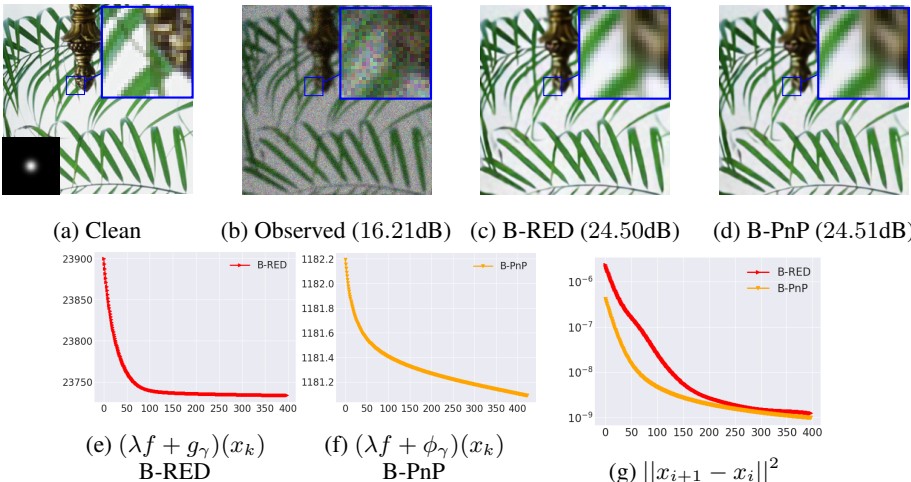

(a) Clean     (b) Observed (16.21dB)     (c) B-RED (24.50dB)     (d) B-PnP (24.51dB)

(e) $(\lambda f + g_\gamma)(x_k)$
B-RED

(f) $(\lambda f + \phi_\gamma)(x_k)$
B-PnP

(g) $||x_{i+1} - x_i||^2$

Figure 3: Deblurring from the indicated Gaussian blur kernel and Poisson noise with $\alpha = 60$.

of the problem, we propose a new deep denoiser, parametrized by $h$, which provably writes as the Bregman proximal operator of a nonconvex potential. We argue that this denoiser should be trained on a particular noise model, called Bregman noise, that also depends on $h$. By plugging this denoiser in the BPG algorithm, we propose two new plug-and-play algorithms, called B-PnP and B-RED, and show that both algorithms converge to stationary points of explicit nonconvex functionals. We apply this framework to Poisson image inverse problem. Experiments on image deblurring illustrate numerically the convergence and the efficiency of the approach.

The central significance of our work stems from its theoretical study but we recognize certain limits within our experimental results. First, when applied to deblurring with Poisson noise, our proposed algorithms do not outperform existing methods in terms of PSNR. Second, while we prove that B-RED is convergent without restriction, the convergence of B-PnP depends on a specific convexity condition. Despite being confirmed with experiments and having robust theoretical foundations, this assumption could potentially be not verified when applied to non-natural images that significantly differ from those in the training dataset. Finally, due to the nonconvex nature of our proposed prior, the practical performance of the algorithms can be sensitive to their initialization.

**Acknowledgements** This work was funded by the French ministry of research through a CDSN grant of ENS Paris-Saclay. It has also been carried out with financial support from the French Research Agency through the PostProdLEAP and Mistic projects (ANR-19-CE23-0027-01 and ANR-19-CE40-005). It has also been supported by the NSF CAREER award under grant CCF-2043134.

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

# A  Definitions

## A.1  Legendre functions, Bregman divergence

**Definition 1** (Legendre function, Rockafellar [1997]). *Let $h : C \subseteq \mathbb{R}^n \to \mathbb{R} \cup \{+\infty\}$ be a proper lower semi-continuous convex function. It is called:*

(i) *essentially smooth, if $h$ is differentiable on $int \, \mathrm{dom}(h)$, with moreover $||\nabla h(x^k)|| \to \infty$ for every sequence $\{x^k\}_{k \in \mathbb{N}}$ of $int \, \mathrm{dom}(h)$ converging towards a boundary point of $\mathrm{dom}(h)$.*

(ii) *of Legendre type if $h$ is essentially smooth and strictly convex on $int \, \mathrm{dom}(h)$.*

We also recall the following properties of Legendre functions that are used without justification in our analysis (see [Rockafellar, 1997, Section 26] for more details):

- $h$ is of Legendre type if and only if its convex conjugate $h^*$ is of Legendre type.
- For $h$ of Legendre type, $dom(\nabla h) = int \, \mathrm{dom}(h)$, $\nabla h$ is a bijection from $int \, \mathrm{dom}(h)$ to $int \, \mathrm{dom}(h^*)$ and $(\nabla h)^{-1} = \nabla h^*$.

**Examples**   We are interested in the two following Legendre functions

- $L_2$ potential : $h(x) = \frac{1}{2}||x||^2$, $C = \mathrm{dom}(h) = \mathrm{dom}(h^*) = \mathbb{R}^n$, $D_h(x, y) = \frac{1}{2}||x - y||^2$
- Burg's entropy : $h(x) = -\sum_{i=1}^{n} \log(x_i)$, $C = \mathrm{dom}(h) = \mathbb{R}^n_{++}$, $dom(h^*) = \mathbb{R}^n_{--}$ and $D_h(x, y) = \sum_{i=1}^{n} \frac{x_i}{y_i} - \log\left(\frac{x_i}{y_i}\right) - 1$.

## A.2  Nonconvex subdifferential

Following Attouch et al. [2013], we use as notion of subdifferential of a proper, nonconvex function $\Phi$ the *limiting subdifferential*

$$\partial\Phi(x) = \left\{\omega \in \mathbb{R}^n, \exists x_k \to x, \Phi(x_k) \to \Phi(x), \omega_k \to \omega, \omega_k \in \hat{\partial}\Phi(x_k)\right\} \tag{33}$$

with $\hat{\partial}\Phi$ the Fréchet subdifferential of $\Phi$. We have

$$\left\{\omega \in \mathbb{R}^n, \exists x_k \to x, \Phi(x_k) \to \Phi(x), \omega_k \to \omega, \omega_k \in \partial\Phi(x_k)\right\} \subseteq \partial\Phi(x). \tag{34}$$

## A.3  Kurdyka-Lojasiewicz (KL) property and subanalytic functions.

**Definition 2** (Kurdyka-Lojasiewicz (KL) [Attouch et al., 2013]). *A function $f : \mathbb{R}^n \longrightarrow \mathbb{R} \cup +\infty$ is said to have the Kurdyka-Lojasiewicz property at $x^* \in dom(f)$ if there exists $\eta \in (0, +\infty)$, a neighborhood $U$ of $x^*$ and a continuous concave function $\psi : [0, \eta) \longrightarrow \mathbb{R}_+$ such that $\psi(0) = 0$, $\psi$ is $\mathcal{C}^1$ on $(0, \eta)$, $\psi' > 0$ on $(0, \eta)$ and $\forall x \in U \cap [f(x^*) < f < f(x^*) + \eta]$, the Kurdyka-Lojasiewicz inequality holds:*

$$\psi'(f(x) - f(x^*))dist(0, \partial f(x)) \geq 1. \tag{35}$$

*Proper lower semicontinuous functions which satisfy the Kurdyka-Lojasiewicz inequality at each point of $dom(\partial f)$ are called KL functions.*

This condition can be interpreted as the fact that, up to a reparameterization, the function is locally sharp around its critical points *i.e.* we can bound its subgradients away from 0. For more details and interpretations, we refer to [Attouch et al., 2010] and [Bolte et al., 2010]. As shown in Bolte et al. [2007], a large class of functions that have the KL-property is given by **continuous subanalytic functions**.

**Definition 3** (Subanalytic [Bolte et al., 2007]).

- *A subset $S$ of $\mathbb{R}^n$ is a semianalytic set if each point of $\mathbb{R}^n$ admits a neighborhood $V$ for which there exists a finite number of real analytic functions $f_{i,j}, g_{i,j} : \mathbb{R}^n \to \mathbb{R}$ such that*

$$S \cap V = \cup_{j=1}^{p} \cap_{i=1}^{q} \{x \in \mathbb{R}^n, f_{i,j} = 0, g_{i,j} < 0\} \tag{36}$$

- A subset $S$ of $\mathbb{R}^n$ is a subanalytic set if each point of $\mathbb{R}^n$ admit a neighborhood $V$ for which

$$S \cap V = \{x \in \mathbb{R}^n, (x, y) \in U\} \tag{37}$$

where $U$ is a bounded semianalytic subset of $\mathbb{R}^n \times \mathbb{R}^p$ with $p \geq 1$.

- A function $f : \mathbb{R}^n \to \mathbb{R} \cup \{+\infty\}$ (resp. $f : \mathbb{R}^n \to \mathbb{R}^m$) is called subanalytic if its graph $\{(x, y) \in \mathbb{R}^n \times \mathbb{R}, y = f(x)\}$ (resp. $\{(x, y) \in \mathbb{R}^n \times \mathbb{R}^m, y = f(x)\}$) is a subanalytic subset of $\mathbb{R}^n \times \mathbb{R}$ (resp. $\mathbb{R}^n \times \mathbb{R}^m$).

Subanalytic functions encompass both analytic functions and semialgebraic [Coste, 2000] functions [Shiota, 2012]. Stability results of subanalytic functions by sum, product, and composition are given in [Shiota, 2012, Section 1.2].

## B   More details on the Bregman Denoising Prior

We detail here the calculations realized in Section 3.

We first verify that the Bregman noise conditional probability belongs to the regular *exponential family of distributions*. Indeed for $T(y) = \gamma \nabla h(y)$, $\psi(x) = \gamma h(x) - \rho(x)$ and $p_0(y) = \exp\left(\gamma h(y) - \gamma \langle \nabla h(y), y \rangle\right)$

$$
\begin{aligned}
p(y|x) &= p_0(y) \exp\left(\langle x, T(y) \rangle - \psi(x)\right) \\
&= \exp\left(\gamma h(y) - \gamma \langle \nabla h(y), y \rangle\right) \exp\left(\gamma \langle x, \nabla h(y) \rangle - \gamma h(x) + \rho(x)\right) \\
&= \exp\left(-\gamma(h(x) - h(y) - \langle \nabla h(y), x - y \rangle) + \rho(x)\right) \\
&= \exp\left(-\gamma D_h(x, y) + \rho(x)\right)
\end{aligned}
\tag{38}
$$

which corresponds to the Bregman noise conditional probability introduced in Equation (6).

The Tweedie formula of the posterior mean is then [Kim and Ye, 2021]

$$\nabla T(y) . \hat{x}_{MMSE}(y) = -\nabla \log p_0(y) + \nabla \log p_Y(y). \tag{39}$$

Using

$$\nabla \log p_0(y) = \gamma \nabla h(y) - \gamma \nabla h(y) - \gamma \nabla^2 h(y) . y = -\gamma \nabla^2 h(y) . y, \tag{40}$$

we get

$$\gamma \nabla^2 h(y) . \hat{x}_{MMSE}(y) = \gamma \nabla^2 h(y) . y + \nabla \log p_Y(y). \tag{41}$$

As $h$ is strictly convex, $\nabla^2 h(y)$ is invertible and

$$\hat{x}_{MMSE}(y) = y + \frac{1}{\gamma} (\nabla^2 h(y))^{-1} . \nabla \log p_Y(y). \tag{42}$$

## C   Proof of Proposition 1

**A new characterization of Bregman proximal operators** We first derive in Proposition 2 an extension of [Gribonval and Nikolova, 2020, Corollary 5 a)] when $h$ is strictly convex and thus $\nabla h$ is invertible. We will then see that Proposition 1 is a direct application of this result.

**Proposition 2.** *Let $h$ of Legendre type on $\mathbb{R}^n$. Let $\zeta : int \, dom(h) \to \mathbb{R}^n$. Suppose $Im(\zeta) \subset dom(h)$. The following properties are equivalent:*

(i) *There is $\phi : \mathbb{R}^n \to \mathbb{R} \cup \{+\infty\}$ such that $Im(\zeta) \subset dom(\phi)$ and for each $y \in int \, dom(h)$*

$$\zeta(y) \in \underset{x \in \mathbb{R}^n}{\arg\min} \{D_h(x, y) + \phi(x)\}. \tag{43}$$

(ii) *There is a l.s.c $\psi : \mathbb{R}^n \to \mathbb{R} \cup \{+\infty\}$ proper convex on $int \, dom(h^*)$ such that $\zeta(\nabla h^*(z)) \in \partial \psi(z)$ for each $z \in int \, dom(h^*)$.*

*When (i) holds, $\psi$ can be chosen given $\phi$ with*

$$\psi(z) := \begin{cases} \langle \zeta(\nabla h^*(z)), z \rangle - h(\zeta(\nabla h^*(z))) - \phi(\zeta(\nabla h^*(z))) & \text{if } z \in int \operatorname{dom}(h^*) \\ +\infty & \text{otherwise.} \end{cases} \quad (44)$$

*When (ii) holds, $\phi$ can be chosen given $\psi$ with*

$$\phi(x) := \begin{cases} \langle \zeta(y), \nabla h(y) \rangle - h(\zeta(y)) - \psi(\nabla h(y)) \text{ for } y \in \zeta^{-1}(x) & \text{if } x \in Im(\zeta) \\ +\infty & \text{otherwise.} \end{cases} \quad (45)$$

**Remark 4.** *[Gribonval and Nikolova, 2020, Corollary 5 a)] with $\mathcal{Y} = int \operatorname{dom}(h)$ states that (i) is equivalent to*

(iii) *There is a l.s.c $g : \mathbb{R}^n \to \mathbb{R} \cup \{+\infty\}$ convex such that $\nabla h(\zeta^{-1}(x)) \in \partial g(x)$ for each $x \in Im(\zeta)$.*

*However (ii) and (iii) are not equivalent. We show here that (ii) implies (iii) but the converse is not true. Let $\psi$ convex defined from (ii). Thanks to Legendre-Fenchel identity, we have that*

$$\forall z \in int \operatorname{dom}(h^*), \ \zeta(\nabla h^*(z)) \in \partial\psi(z) \quad (46)$$
$$\Leftrightarrow \forall z \in int \operatorname{dom}(h^*), \ z \in \partial\psi^*(\zeta(\nabla h^*(z))) \quad (47)$$
$$\Leftrightarrow \forall y \in int \operatorname{dom}(h), \ \nabla h(y) \in \partial\psi^*(\zeta(y)) \quad (48)$$
$$\Rightarrow \forall x \in Im(\zeta), \ \nabla h(\zeta^{-1}(x)) \in \partial\psi^*(x). \quad (49)$$

*Therefore (ii) implies (iii) with $g = \psi^*$. However, the last lign is just an implication.*

*Proof.* We follow the same order of arguments from [Gribonval and Nikolova, 2020]. We first prove a general result reminiscent to [Gribonval and Nikolova, 2020, Theorem 3] for a general form of divergence function and then apply this result to Bregman divergences.

**Lemma 1.** *Let $a : \mathcal{Y} \subseteq \mathbb{R}^n \to \mathbb{R} \cup \{+\infty\}$, $b : \mathbb{R}^n \to \mathbb{R} \cup \{+\infty\}$, $A : \mathcal{Y} \to \mathcal{Z}$ bijection from $\mathcal{Y}$ to $\mathcal{Z}$ (with $\mathcal{Z} \subset \mathbb{R}^n$). Consider $\zeta : \mathcal{Y} \to \mathbb{R}^n$. Let $D(x,y) := a(y) - \langle x, A(y) \rangle + b(x)$. Suppose $\operatorname{Im}(\zeta) \subset \operatorname{dom}(b)$. The following properties are equivalent:*

(i) *There is $\phi : \mathbb{R}^n \to \mathbb{R} \cup \{+\infty\}$ such that $\operatorname{Im}(\zeta) \subset \operatorname{dom}(\phi)$ and for each $y \in \mathcal{Y}$*

$$\zeta(y) \in \underset{x \in \mathbb{R}^n}{\arg\min}\{D(x,y) + \phi(x)\}. \quad (50)$$

(ii) *There is a l.s.c $\psi : \mathbb{R}^n \to \mathbb{R} \cup \{+\infty\}$ proper convex such that $\zeta(A^{-1}(z)) \in \partial\psi(z)$ for each $z \in \mathcal{Z}$.*

*When (i) holds, $\psi$ can be chosen given $\phi$ with*

$$\psi(z) := \begin{cases} \langle \zeta(A^{-1}(z)), z \rangle - b(\zeta(A^{-1}(z))) - \phi(\zeta(A^{-1}(z))) & \text{if } z \in \mathcal{Z} \\ +\infty & \text{otherwise.} \end{cases} \quad (51)$$

*When (ii) holds, $\phi$ can be chosen given $\psi$ with*

$$\phi(x) := \begin{cases} \langle \zeta(y), A(y) \rangle - b(\zeta(y)) - \psi(A(y)) \text{ for } y \in \zeta^{-1}(x) & \text{if } x \in Im(\zeta) \\ +\infty & \text{otherwise.} \end{cases} \quad (52)$$

Before proving the Lemma, we can directly see that Proposition 2 is the specialization of Lemma 1 with Bregman divergences. Given $h$ of Legendre-type, the divergence $D(x,y) = a(y) - \langle x, A(y) \rangle + b(x)$ becomes the Bregman divergence $D_h(x,y)$ defined in (5) for $\mathcal{Y} = int \operatorname{dom}(h)$, $\mathcal{Z} = int \operatorname{dom}(h^*)$, $A(y) := \nabla h(y)$, $a(y) := \langle \nabla h(y), y \rangle - h(y)$ and $b(x) := h(x)$ if $x \in \operatorname{dom}(h)$ and $+\infty$ otherwise. For $h$ of Legendre-type, $\nabla h$ is a bijection from $int \operatorname{dom}(h)$ to $int \operatorname{dom}(h^*)$ and $(\nabla h)^{-1} = \nabla h^*$. $\qquad\square$

*Proof.* We now prove Lemma 1. We follow the same arguments as the proof of [Gribonval and Nikolova, 2020, Theorem 3(c)]. **(i)** $\Rightarrow$ **(ii) :**
Define

$$\rho(z) = \begin{cases} \langle \zeta(A^{-1}(z)), z \rangle - b(\zeta(A^{-1}(z))) - \phi(\zeta(A^{-1}(z))) & \text{if } z \in \mathcal{Z} \\ +\infty & \text{else.} \end{cases} \tag{53}$$

As we assume $\text{Im}(\zeta) \subset \text{dom}(b)$ and $\text{Im}(\zeta) \subset \text{dom}(\psi)$, we have $\rho : \mathbb{R}^n \to \mathbb{R} \cup \{+\infty\}$ and $\text{dom}(\rho) = \mathcal{Z}$. Let $z \in \mathcal{Z}$ and $y = A^{-1}(z)$. From (i), $\zeta(y)$ is the global minimizer of $x \to D(x, y) + \phi(x)$ or of $x \to -\langle x, A(y) \rangle + b(x) + \phi(x)$ and $\forall z' \in \mathcal{Z}, y' = A^{-1}(z')$,

$$\begin{aligned} \rho(z') - \rho(z) &= \langle \zeta(A^{-1}(z')), z' \rangle - b(\zeta(A^{-1}(z'))) - \phi(\zeta(A^{-1}(z'))) \\ &\quad - \langle \zeta(A^{-1}(z)), z \rangle + b(\zeta(A^{-1}(z))) + \phi(\zeta(A^{-1}(z))) \\ &= \langle \zeta(y'), A(y') \rangle - b(\zeta(y')) - \phi(\zeta(y')) - \langle \zeta(y), A(y) \rangle + b(\zeta(y)) + \phi(\zeta(y)) \\ &= \langle \zeta(y), A(y') - A(y) \rangle + \langle \zeta(y'), A(y') \rangle - b(\zeta(y')) - \phi(\zeta(y')) \\ &\quad - \langle \zeta(y), A(y') \rangle + b(\zeta(y)) + \phi(\zeta(y)) \\ &\geq \langle \zeta(y), A(y') - A(y) \rangle = \langle \zeta(A^{-1}(z)), z' - z \rangle. \end{aligned} \tag{54}$$

By definition of the subdifferential, this shows that

$$\zeta(A^{-1}(z)) \in \partial \rho(z). \tag{55}$$

Let $\tilde{\rho}$ the lower convex envelope of $\rho$ (pointwise supremum of all the convex l.s.c functions below $\rho$). $\tilde{\rho}$ is proper convex l.s.c. and $\forall z \in \mathcal{Z}, \partial \rho(z) \neq \emptyset$. By [Gribonval and Nikolova, 2020, Proposition 3], $\forall z \in \mathcal{Z}, \rho(z) = \tilde{\rho}(z)$ and $\partial \rho(z) = \partial \tilde{\rho}(z)$. Thus for $\psi = \tilde{\rho}$, we get (ii).

**(ii)** $\Rightarrow$ **(i) :**
Define $\eta : \mathcal{Y} \to \mathbb{R}$ by

$$\eta(y) := \langle \zeta(y), A(y) \rangle - \psi(A(y)). \tag{56}$$

The previous definition is valid because by (ii), $\text{Im}(A) = \mathcal{Z} \subset \text{dom}(\partial \psi) \subset \text{dom}(\psi)$ and therefore $\psi(A(y)) < +\infty$. By (ii), $\forall z, z' \in \mathcal{Z}$,

$$\psi(z) - \psi(z') \geq \langle \zeta(A^{-1}(z')), z - z' \rangle \tag{57}$$

which gives $\forall y, y' \in \mathcal{Y}$,

$$\psi(A(y)) - \psi(A(y')) \geq \langle \zeta(y'), A(y) - A(y') \rangle. \tag{58}$$

This yields

$$\begin{aligned} \eta(y') - \eta(y) &= \langle \zeta(y'), A(y') \rangle - \psi(A(y')) - \langle \zeta(y), A(y) \rangle + \psi(A(y)) \\ &\geq \langle \zeta(y') - \zeta(y), A(y) \rangle. \end{aligned} \tag{59}$$

We define $\theta : \mathbb{R}^n \to \mathbb{R} \cup \{+\infty\}$ obeying $\text{dom}(\theta) = \text{Im}(\zeta)$ with

$$\theta(x) := \begin{cases} \eta(y) \text{ for } y \in \zeta^{-1}(x) & \text{if } x \in Im(\zeta) \\ +\infty & \text{otherwise.} \end{cases} \tag{60}$$

For $y, y' \in \zeta^{-1}(x)$, as $\zeta(y') = \zeta(y)$, we have by (59) $\eta(y') - \eta(y) \geq 0$ and $\eta(y) - \eta(y') \geq 0$ and thus $\eta(y') = \eta(y)$. The definition of $\theta$ is thus independent of the choice of $y \in \zeta^{-1}(x)$.

For $x' \in Im(\zeta)$, $x' = \zeta(y')$. Using the previous inequality with $\eta$, we get

$$\begin{aligned} \theta(x') - \theta(\zeta(y)) &= \theta(\zeta(y')) - \theta(\zeta(y)) \\ &= \eta(y') - \eta(y) \\ &\geq \langle \zeta(y') - \zeta(y), A(y) \rangle \\ &= \langle x' - \zeta(y), A(y) \rangle, \end{aligned} \tag{61}$$

that is to say, $\forall x' \in Im(\zeta)$

$$\theta(x') - \langle x', A(y) \rangle \geq \theta(\zeta(y)) - \langle \zeta(y), A(y) \rangle. \tag{62}$$

Given the definition of $\theta$, this is also true for $x' \notin Im(\zeta)$.

We set $\phi = \theta - b$. As $\text{Im}(\zeta) \subset \text{dom}(b)$, $b(\zeta(y)) < +\infty$ and $\phi : \mathbb{R}^n \to \mathbb{R} \cup \{+\infty\}$. Adding $a(y)$ on both sides, we get

$$\forall x' \in \mathbb{R}^n, \ b(x') + \phi(x') - \langle x', A(y) \rangle + a(y) \geq b(\zeta(y)) + \phi(\zeta(y)) - \langle \zeta(y), A(y) \rangle + a(y).$$

$$\Leftrightarrow \ \forall x' \in \mathbb{R}^n, \ \phi(x') + D(x', y) \geq \phi(\zeta(y)) + D(\zeta(y), y) \tag{63}$$

$$\Leftrightarrow \ \zeta(y) \in \arg\min_x \phi(x) + D(x, y)$$

As this is true for all $y \in \mathcal{Y}$, we get the desired result. $\qquad\square$

Eventually, Proposition 1 is a direct application of Proposition 2 with $\zeta = \mathcal{B}_\gamma : \mathbb{R}^n \to \mathbb{R}^n$ defined on $int\,\text{dom}(h)$ by

$$\mathcal{B}_\gamma(y) = \nabla(\psi_\gamma \circ \nabla h^*) \circ \nabla h(y). \tag{64}$$

The function $\mathcal{B}_\gamma$ verifies $\forall z \in int\,\text{dom}(h^*)$,

$$\mathcal{B}_\gamma(\nabla h^*(z)) = \nabla(\psi_\gamma \circ \nabla h^*)(z) \tag{65}$$

and $\psi = \psi_\gamma \circ \nabla h^*$ is assumed convex on $int\,\text{dom}(h^*)$. From Proposition 2, we get that there is $\phi_\gamma : \mathbb{R}^n \to \mathbb{R} \cup \{+\infty\}$ such that for each $y \in int\,\text{dom}(h)$

$$\mathcal{B}_\gamma(y) \in \arg\min\{D_h(x, y) + \phi_\gamma(x)\}. \tag{66}$$

Moreover, for $y \in int\,\text{dom}(h)$,

$$\mathcal{B}_\gamma(y) = \nabla(\psi_\gamma \circ \nabla h^*) \circ \nabla h(y) \tag{67}$$

$$= \nabla^2 h^*(\nabla h(y)).\nabla\psi_\gamma \circ \nabla h^* \circ \nabla h(y) \tag{68}$$

$$= \nabla^2 h^*(\nabla h(y)).\nabla\psi_\gamma(y). \tag{69}$$

As $h$ is assumed strictly convex on $int\,\text{dom}(h)$, for $y \in int\,\text{dom}(h)$, the Hessian of $h$, denoted as $\nabla^2 h(y)$ is invertible. By differentiating

$$\nabla h^*(\nabla h(y)) = y \tag{70}$$

we get

$$\nabla^2 h^*(\nabla h(y)) = (\nabla^2 h(y))^{-1}, \tag{71}$$

so that

$$\mathcal{B}_\gamma(y) = (\nabla^2 h(y))^{-1}.\nabla\psi_\gamma(y). \tag{72}$$

With the definition (11), we directly get

$$\mathcal{B}_\gamma(y) = (\nabla^2 h(y))^{-1}.\nabla\psi_\gamma(y) = y - (\nabla^2 h(y))^{-1}.\nabla g_\gamma(y). \tag{73}$$

Finally, Proposition 2 also indicates that $\phi_\gamma$ can be chosen given $\psi_\gamma$ with

$$\phi_\gamma(x) := \begin{cases} \langle \mathcal{B}_\gamma(y), \nabla h(y) \rangle - h(\mathcal{B}_\gamma(y)) - \psi_\gamma \circ \nabla h^*(\nabla h(y)) \ \text{for } y \in \mathcal{B}_\gamma^{-1}(x) & \text{if } x \in Im(\mathcal{B}_\gamma) \\ +\infty & \text{otherwise.} \end{cases} \tag{74}$$

This gives $\forall y \in int\,\text{dom}(h)$,

$$\begin{aligned} \phi_\gamma(\mathcal{B}_\gamma(y)) &= \langle \mathcal{B}_\gamma(y), \nabla h(y) \rangle - h(\mathcal{B}_\gamma(y)) - \psi_\gamma \circ \nabla h^*(\nabla h(y)) \\ &= \langle \mathcal{B}_\gamma(y) - y, \nabla h(y) \rangle - h(\mathcal{B}_\gamma(y)) + h(y) + \langle y, \nabla h(y) \rangle - h(y) - \psi_\gamma(y) \\ &= -D_h(\mathcal{B}_\gamma(y), y) + \langle y, \nabla h(y) \rangle - h(y) - \psi_\gamma(y) \\ &= -D_h(\mathcal{B}_\gamma(y), y) + g_\gamma(y). \end{aligned} \tag{75}$$

# D The Bregman Proximal Gradient (BPG) algorithm

## D.1 Convergence analysis of the nonconvex BPG algorithm

We study in this section the convergence of the BPG algorithm

$$x^{k+1} \in T_\tau(x_k) = \arg\min_{x \in \mathbb{R}^n}\{\mathcal{R}(x) + \langle x - x^k, \nabla F(x^k) \rangle + \frac{1}{\tau}D_h(x, x^k)\}. \tag{76}$$

for minimizing $\Psi = F + \mathcal{R}$ with nonconvex functions $F$ and/or $\mathcal{R}$.

For the rest of the section, we take the following general assumptions.

**Assumption 3.**

  (i) $h : \mathbb{R}^n \to \mathbb{R}$ *is of Legendre-type.*

  (ii) $F : \mathbb{R}^n \to \mathbb{R}$ *is proper,* $\mathcal{C}^1$ *on* $int \operatorname{dom}(h)$*, with* $\operatorname{dom}(h) \subset \operatorname{dom}(F)$*.*

  (iii) $\mathcal{R} : \mathbb{R}^n \to \mathbb{R}$ *is proper, lower semi-continuous with* $\operatorname{dom} \mathcal{R} \cap int \operatorname{dom}(h) \neq \emptyset$*.*

  (iv) $\Psi = F + \mathcal{R}$ *is lower-bounded, coercive and verifies the Kurdyka-Lojasiewicz (KL) property (defined in Appendix A.3).*

  (v) *For* $x \in int \operatorname{dom}(h)$*,* $T_\tau(x)$ *is nonempty and included in* $int \operatorname{dom}(h)$*.*

Note that, since $\mathcal{R}$ is nonconvex, the mapping $T_\tau$ is not in general single-valued.

Assumption (v) is required for the algorithm to be well-posed. As shown in [Bauschke et al., 2017, Bolte et al., 2018], one sufficient condition for $T_\tau(x) \neq \emptyset$ is the supercoercivity of function $h + \lambda \mathcal{R}$ for all $\lambda > 0$, that is $\lim_{||x|| \to +\infty} \frac{h(x) + \lambda \mathcal{R}(x)}{||x||} = +\infty$. As $T_\tau(x) \subset \operatorname{dom}(h)$, $T_\tau(x) \subset int \operatorname{dom}(h)$ is true when $\operatorname{dom}(h)$ is open (which is the case for Burg's entropy for example).

The convergence of the BPG algorithm in the nonconvex setting is studied by the authors of [Bolte et al., 2018]. Under the main assumption that $Lh - F$ is convex on $int \operatorname{dom}(h)$, they show first the sufficient decrease property (and thus convergence) of the function values, and second, global convergence of the iterates. However, as we will develop in Appendix D.3, $Lh - F$ is not convex on the full domain of $int \operatorname{dom}(h)$ but only on the compact subset $\operatorname{dom}(\mathcal{R})$.

One can verify that all the iterates (76) belong to the convex set

$$T_\tau(x) \subset \operatorname{Conv}(\operatorname{dom} \mathcal{R}) \cap int \operatorname{dom}(h), \tag{77}$$

where $\operatorname{Conv}(E)$ stands for the convex envelope of $E$. We argue that it is enough to assume $Lh - F$ convex on this convex subset with the following assumption.

**Assumption 4.** *There is* $L > 0$ *such that,* $Lh - F$ *is convex on* $\operatorname{Conv}(\operatorname{dom} \mathcal{R}) \cap int \operatorname{dom}(h)$*.*

We can now prove a result similar than [Bolte et al., 2018, Proposition 4.1].

**Proposition 3.** *Under Assumptions 3 and 4, let* $(x^k)_{k \in \mathbb{N}}$ *be a sequence generated by* (18) *with* $0 < \tau L < 1$*. Then the following properties hold*

  (i) $(\Psi(x^k))_{k \in \mathbb{N}}$ *is non-increasing and converges.*

  (ii) $\sum_k D_h(x^{k+1}, x^k) < \infty$ *and* $\min_{0 \leq k \leq K} D_h(x^{k+1}, x^k) = O(1/K)$*.*

*Proof.* We adapt here the proof from [Bolte et al., 2018] to the case where $Lh - F$ is not globally convex but only convex on the convex subset $\operatorname{Conv}(\operatorname{dom} \mathcal{R}) \cap int \operatorname{dom}(h)$.

**Sufficient decrease property**   We first show that the sufficient decrease property of $\Psi(x_k)$ holds. This is true because the following characterisation of $\mathcal{C}^1$ convex functions holds on a convex subset. We recall this classical proof for the sake of completeness.

**Lemma 2.** *Let* $f : \mathcal{X} \to \mathbb{R}^n$ *be of class* $\mathcal{C}^1$*, then* $f$ *is locally convex on* $C$ *a convex subset of* $\mathcal{X} = dom(f)$ *if and only if* $\forall x, y \in C$*,* $D_f(x, y) = f(x) - f(y) - \langle \nabla f(y), x - y \rangle \geq 0$*.*

*Proof.* $\Rightarrow$ Let $x, y \in C$, for all $t \in (0, 1)$, $x + t(y - x) \in C$ and by convexity of $f$

$$f(y + t(x - y)) \leq f(y) + t(f(x) - f(y)), \tag{78}$$

i.e.

$$\frac{f(y + t(x - y)) - f(y)}{t} \leq f(x) - f(y) \tag{79}$$

and

$$\langle \nabla f(y), x - y \rangle = \lim_{t \to 0^+} \frac{f(y + t(x - y)) - f(y)}{t} \leq f(x) - f(y). \tag{80}$$

$\Leftarrow$ Let $x, y \in C$, $t \in (0, 1)$ and $z = y + t(x - y) \in C$. We have

$$f(y) \geq f(z) + \langle \nabla f(z), y - z \rangle \tag{81}$$
$$f(x) \geq f(z) + \langle \nabla f(z), x - z \rangle. \tag{82}$$

Combining both equations gives

$$
\begin{aligned}
tf(x) + (1 - t)f(y) &\leq t(f(z) + \langle \nabla f(z), x - z \rangle) + (1 - t)(f(z) + \langle \nabla f(z), y - z \rangle) \\
&= f(z) + \langle \nabla f(z), tx + (1 - t)y - z \rangle \\
&= f(tx + (1 - t)y).
\end{aligned} \tag{83}
$$

$\square$

Therefore, we have $Lh - F$ convex on $C$, if and only if, $\forall x, y \in C$, $D_{Lh-F}(x, y) \geq 0$, *i.e.* $D_F(x, y) \leq LD_h(x, y)$.

The rest of the proof is identical to the one of [Bolte et al., 2018] and we recall it here. Given the optimality conditions in (76), all the iterates $x_k \in C$ satisfy

$$\mathcal{R}(x^{k+1}) + \langle x^{k+1} - x^k, \nabla F(x_k) \rangle + \frac{1}{\tau} D_h(x^{k+1}, x^k) \leq \mathcal{R}(x^k) \tag{84}$$

using $D_F(x, y) \leq LD_h(x, y)$, we get

$$
\begin{aligned}
(\mathcal{R}(x^{k+1}) + F(x^{k+1})) - (\mathcal{R}(x^k) + F(x^k)) &\leq -\frac{1}{\tau} D_h(x^{k+1}, x^k) + LD_h(x^{k+1}, x^k) \\
&= (L - \frac{1}{\tau}) D_h(x^{k+1}, x^k) \leq 0
\end{aligned} \tag{85}
$$

which, together with the fact that $\Psi$ is lower bounded, proves (i). Summing the previous inequality from $k = 0$ to $K - 1$ gives

$$0 \leq \sum_{k=0}^{K-1} D_h(x^{k+1}, x^k) \leq \frac{\tau}{1 - \tau L}(\Psi(x_0) - \Psi(x_K)) \leq \frac{\tau}{1 - \tau L}(\Psi(x_0) - \inf_{x \in C} \Psi(x)) < +\infty. \tag{86}$$

Thus $(D_h(x^{k+1}, x^k))_k$ is summable and converges to $0$ when $k \to +\infty$. Finally

$$\min_{0 \leq k \leq K} D_h(x^{k+1}, x^k) \leq \frac{1}{K+1} \sum_{k=0}^{K} D_h(x^{k+1}, x^k) \leq \frac{1}{K+1} \frac{\tau}{1 - \tau L}(\Psi(x_0) - \inf_{x \in C} \Psi(x)). \tag{87}$$

$\square$

To prove global convergence of the iterates upon the Kurdyka-Lojasiewicz (KL) property, [Bolte et al., 2018, Theorem 4.1] is based on the hypotheses (a) $\text{dom}(h) = \mathbb{R}^n$ and $h$ is strongly convex on $\mathbb{R}^n$ and (b) $\nabla h$ and $\nabla F$ are Lipschitz continuous on any bounded subset of $\mathbb{R}^n$. These assumptions are clearly not verified for $h$ being the Burg's entropy (22) or $F$ the Poisson data-fidelity term (2). Indeed, in that case, $\text{dom}(h) = \mathbb{R}_{++}^n$, and $h$ is strongly convex only on bounded sets. Moreover $F$ and $h$ are not Lipschitz continuous near $0$.

However, thanks to the proven decrease of the iterates and as $\Psi$ is assumed coercive, the iterates remain bounded. We can adopt the following weaker assumptions to ensure that the iterates do not tend to $+\infty$ or $0$.

**Assumption 5.**

*(i) $h$ is strongly convex on any bounded convex subset of its domain.*

*(ii) For all $\alpha > 0$, $\nabla h$ and $\nabla F$ are Lipschitz continuous on $\{\Psi(x) \leq \alpha\}$.*

Under these assumptions, we prove the equivalent of [Bolte et al., 2018, Theorem 4.1].

**Theorem 3.** *Under Assumption 3, 4 and 5, the sequence $(x^k)_{k \in \mathbb{N}}$ generated by (18) with $0 < \tau L < 1$ converges to a critical point of $\Psi$.*

We follow the proof given in [Bolte et al., 2018] with some updates to adapt to our weaker Assumption 5. In particular, we show in Lemma 3 that the sequence $(x_k)_{k\geq 1}$ is still "gradient-like" *i.e.* it verifies the assumptions H1, H2 and H3 from [Attouch et al., 2013]. Once these conditions are verified, the result directly follows from [Bolte et al., 2018, Theorem 6.2].

Let us first note that by coercivity of $\Psi$ and decrease of the iterates $\Psi(x_k)$ (see Proposition 3), the iterates remain in the set

$$\forall k \geq 1, \ \ x_k \in C(x_0) = \{x \in \mathrm{dom}(h), \Psi(x) < \Psi(x_0)\}. \tag{88}$$

**Lemma 3.** *The sequence $(x_k)_{k\geq 1}$ satisfies the following three conditions:*

    **H1 (Sufficient decrease condition)**

$$\Psi(x_k) - \Psi(x_{k+1}) \geq a||x_{k+1} - x_k||^2, \tag{89}$$

    **H2 (Relative error condition)** $\forall k \geq 1$, *there exists $\omega_{k+1} \in \partial\Psi(x_{k+1})$ such that*

$$||\omega_{k+1}|| \leq b||x_{k+1} - x_k||, \tag{90}$$

    **H3 (Continuity condition)** *Any subsequence $(x^{k_i})$ converging towards $x^*$ verifies*

$$\Psi(x^{k_i}) \to \Psi(x^*). \tag{91}$$

*Proof.* **H1 : Sufficient decrease condition**. From (85), we get

$$\Psi(x_k) - \Psi(x_{k+1}) \geq \left(\frac{1}{\tau} - L\right) D_h(x^{k+1}, x^k). \tag{92}$$

Besides, $h$ is assumed to be strongly convex on any bounded convex subset of its domain. Furthermore, notice that $\mathrm{Conv}(C(x_0)) \cap \mathrm{dom}(h)$ is a convex subset of $\mathrm{dom}(h)$ as intersection of convex sets. Therefore, there is $\sigma_h > 0$ such that

$$\forall x, y \in \mathrm{Conv}(C(x_0)) \cap \mathrm{dom}(h), \ \ D_h(x,y) \geq \sigma_h||x - y||^2. \tag{93}$$

With the convention $D_h(x,y) = +\infty$ if $x \notin \mathrm{dom}(h)$ or $y \notin int\,\mathrm{dom}(h)$,

$$\forall x, y \in \mathrm{Conv}(C(x_0)), \ \ D_h(x,y) \geq \sigma_h||x - y||^2. \tag{94}$$

As $\forall k \geq 1, x_k \in C(x_0)$, we get

$$\Psi(x_k) - \Psi(x_{k+1}) \geq \sigma_h \left(\frac{1}{\tau} - L\right) ||x_{k+1} - x_k||^2, \tag{95}$$

which proves (H1).

**H2 : Relative error condition**. Given (76), the optimality condition for the update of $x_{k+1}$ is

$$0 \in \partial\mathcal{R}(x^{k+1}) + \nabla F(x^k) + \frac{1}{\tau}(\nabla h(x^{k+1}) - \nabla h(x^k)). \tag{96}$$

For

$$\omega_{k+1} = \nabla F(x^{k+1}) - \nabla F(x^k) + \frac{1}{\tau}(\nabla h(x^k) - \nabla h(x^{k+1})) \tag{97}$$

we have

$$\omega_{k+1} \in \partial\Psi(x_{k+1}) = \partial\mathcal{R}(x^{k+1}) + \nabla F(x^{k+1}) \tag{98}$$

and

$$||\omega_{k+1}|| \leq ||\nabla F(x^{k+1}) - \nabla F(x^k)|| + \frac{1}{\tau}||\nabla h(x^k) - \nabla h(x^{k+1})||. \tag{99}$$

By assumption, $\nabla F$ and $\nabla h$ are Lipschitz continuous on $C(x_0) = \{\Psi(x) < \Psi(x_0)\}$. As seen before, $\forall k \geq 1, x_k \in C(x_0)$. Thus, there is $b > 0$ such that

$$||\omega_{k+1}|| \leq ||\nabla F(x^{k+1}) - \nabla F(x^k)|| + \frac{1}{\tau}||\nabla h(x^k) - \nabla h(x^{k+1})|| \leq b||x^{k+1} - x^k||. \tag{100}$$

**H3 : Continuity condition**. Let a subsequence $(x^{k_i})$ converging towards $x^*$. Using the optimality in the update of $x_k$ we have

$$\mathcal{R}(x^k) + \langle x^k - x^{k-1}, \nabla F(x^{k-1}) \rangle + \frac{1}{\tau} D_h(x^k, x^{k-1}) \tag{101}$$

$$\leq \mathcal{R}(x^*) + \langle x^* - x^{k-1}, \nabla F(x^{k-1}) \rangle + \frac{1}{\tau} D_h(x^*, x^{k-1}) \tag{102}$$

$$\Leftrightarrow \mathcal{R}(x^k) \leq \mathcal{R}(x^*) + \langle x^* - x^{k-1}, \nabla F(x^{k-1}) \rangle + \frac{1}{\tau} D_h(x^*, x^{k-1}) - \frac{1}{\tau} D_h(x^k, x^{k-1}). \tag{103}$$

From (95) and the fact that $(\Psi(x_k))_k$ converges, we have that $||x^k - x^{k-1}|| \to 0$. Thus $(x^{k_i-1})_i$ also converges towards $x^*$. In addition, since $h$ is continuously differentiable, $D_h(x^*, x^{k-1}) = h(x^*) - h(x^{k-1}) - \langle \nabla h(x^{k-1}), x^* - x^{k-1} \rangle \to 0$. Passing to the limit in (103), we get

$$\limsup_{i \to +\infty} \mathcal{R}(x^{k_i}) \leq \mathcal{R}(x^*). \tag{104}$$

By lower semicontinuity of $\mathcal{R}$ and continuity of $F$, we get the desired result:

$$\mathcal{R}(x^{k_i}) + F(x^{k_i}) \to \mathcal{R}(x^*) + F(x^*). \tag{105}$$

$\square$

**Backtracking** The convergence actually requires to control the NoLip constant. In order to avoid small stepsizes, we adapt the backtracking strategy of [Beck, 2017, Chapter 10] to the BPG algorithm.

Given $\gamma \in (0, 1)$, $\eta \in [0, 1)$ and an initial stepsize $\tau_0 > 0$, the following backtracking update rule on $\tau$ is applied at each iteration $k$:

$$\text{while} \quad \Psi(x_k) - \Psi(T_\tau(x_k)) < \frac{\gamma}{\tau} D_h(T_\tau(x_k), x_k), \quad \tau \longleftarrow \eta\tau. \tag{106}$$

**Proposition 4.** *At each iteration of the algorithm, the backtracking procedure (106) is finite and with backtracking, the convergence results of Proposition 3 and Theorem 3 still hold.*

*Proof.* For a given stepsize $\tau$, we showed in equation (85) that

$$\Phi(x_k) - \Phi(T_\tau(x_k)) \geq \left( \frac{1}{\tau} - L \right) D_h(T_\tau(x_k), x_k). \tag{107}$$

Taking $\tau < \frac{1-\gamma}{L}$, we get $\frac{1}{\tau} - L > \frac{\gamma}{\tau}$ so that

$$\Phi(x_k) - \Phi(T_\tau(x_k)) > \frac{\gamma}{\tau} D_h(T_\tau(x_k), x_k). \tag{108}$$

Hence, when $\tau < \frac{1-\gamma}{L}$, the sufficient decrease condition is satisfied and the backtracking procedure ($\tau \longleftarrow \eta\tau$) must end.

Replacing the former sufficient decrease condition (107) with (108), the rest of the proofs from Proposition 3 and Theorem 3 are identical. $\square$

### D.2 Proof of Theorem 1

*Proof.* We recall the B-RED algorithm

**(B-RED)** $\quad x^{k+1} \in T_\tau(x_k) = \arg\min_{x \in \mathbb{R}^n} \{ i_C(x) + \langle x - x^k, \nabla F_{\lambda,\gamma}(x^k) \rangle + \frac{1}{\tau} D_h(x, x^k) \}.$ (109)

It corresponds to the BPG algorithm (14) with $F = F_{\lambda,\gamma} = \lambda f + g_\gamma$ and $\mathcal{R} = i_C$.

Theorem 1 is a direct application of Proposition 4 that is to say of the convergence results of Proposition 3 and Theorem 3 with backtracking. Given Assumptions 1 and 2, we verify that Assumptions 3, 4 and 5 are verified:

*Assumption 3.* $\mathcal{R} = i_C$ verifies $\text{dom}(\mathcal{R}) \cap \text{int dom}(h) = C \cap \text{int dom}(h) \neq \emptyset$. Moreover, $\mathcal{R}$ is semi-algebraic and thus subanalytic as the indicator function of a closed semi-algebraic set. $g_\gamma$ and $f$

being both assumed subanalytic and lower-bounded, by [Shiota, 2012, I.2.1.9], their sum $\lambda f + g_\sigma$ is then subanalytic (up to adding a constant to make $f$ and $g_\sigma$ non-negative). $\Psi = F_{\lambda,\gamma} + i_C$ is then also subanalytic, and thus KL. $\Psi$ is also lower-bounded and coercive as $f$ is lower-bounded and coercive and $g_\gamma$ is lower-bounded. Finally, for $x \in \text{int dom}(h)$, $T_\tau(x)$ is non-empty as $h + \lambda i_C$ is supercoercive.

*Assumption 4.* By summing convex functions, using Assumption 1(iii) and Assumption 2(ii), $L = \lambda L_f + L_\gamma$ verifies $Lh - (\lambda f + g_\gamma)$ convex on $\text{Conv}(\text{dom } \mathcal{R}) \cap int\, \text{dom}(h) = C \cap int\, \text{dom}(h)$.

*Assumption 5.* As $g_\gamma$ is assumed to have globally Lipschitz continuous gradient, this follows directly from Assumption 1(iv). $\qquad\square$

### D.3 Proof of Theorem 2

$$\textbf{(B-PnP)} \quad x^{k+1} = \mathcal{B}_\gamma \circ \nabla h^*(\nabla h - \lambda \nabla f)(x_k). \tag{110}$$

It corresponds to the BPG algorithm (14) with $F = \lambda f$ and $\mathcal{R} = \phi_\gamma$.

Theorem 2 is a direct application of the convergence results of Proposition 3 and Theorem 3. We now denote $\Psi = \lambda f + \psi_\gamma$. Given Assumptions 1, we verify that Assumptions 3, 4 and 5 are verified.

*Assumption 3.* For $\mathcal{R} = \phi_\gamma$, we have $Im(\mathcal{B}_\gamma) \subset \text{dom}(\phi_\gamma)$ and as for $y \in \text{int dom}(h)$, $Im(\mathcal{B}_\gamma) \subset \text{int dom}(h)$ (by equation 13), we get $\text{dom}(\mathcal{R}) \cap \text{int dom}(h) \neq \emptyset$. $\phi_\gamma$ and $f$ being both assumed subanalytic and lower-bounded, by [Shiota, 2012, I.2.1.9], their sum $\lambda f + \phi_\sigma$ is then subanalytic (up to adding a constant to make $f$ and $\phi_\sigma$ non-negative) and thus KL. $\Psi$ is coercive because $f$ is coercive and $\phi_\gamma$ is lower-bounded. Finally, $T_\tau(x)$ well-posed is ensured via Proposition 1.

*Assumption 4.* This is the L-smad property of $f$ given by Assumption 1.

*Assumption 5.* This is directly given by Assumption 1(iv).

## E  Application to Poisson Inverse Problems

### E.1  Burg's entropy Bregman noise model

For $h$ being the Burg's entropy (22), the Bregman noise model (6) writes for $x, y \in \mathbb{R}^n_{++}$ as

$$
\begin{aligned}
p(y|x) &= \exp\left(-\gamma D_h(x,y) + \rho(x)\right) \\
&= \exp(\rho(x)) \exp\left(-\gamma(h(x) - h(y) - \langle \nabla h(y), x - y \rangle)\right) \\
&= \exp(\rho(x)) \exp\left(-\gamma\Big(\sum_{i=1}^n -\log(x_i) + \log(y_i) - 1 + \frac{x_i}{y_i}\Big)\right) \\
&= \exp(\rho(x) + n\gamma) \prod_{i=1}^n \left(\frac{x_i}{y_i}\right)^\gamma \exp\left(-\gamma\frac{x_i}{y_i}\right).
\end{aligned}
\tag{111}
$$

### E.2  Inverse Gamma Bregman denoiser

**Derivation of** (29)   We first derive here the condition for convexity of $\eta_\gamma := \psi_\gamma \circ \nabla h^*$ introduced in Section 5.1.

We have

$$\nabla \eta_\gamma(x) = \nabla^2 h^*(x).\nabla \psi_\gamma(\nabla h^*(x)) \tag{112}$$

and

$$\nabla^2 \eta_\gamma(x) = (\nabla^2 h^*(x))^2.\nabla^2 \psi_\gamma(\nabla h^*(x)) + \nabla^3 h^*(x).\nabla \psi_\gamma(\nabla h^*(x)) \tag{113}$$

which gives

$$\nabla^2 \eta_\gamma(\nabla h(y)) = (\nabla^2 h^*(\nabla h(y)))^2.\nabla^2 \psi_\gamma(y) + \nabla^3 h^*(\nabla h(y)).\nabla \psi_\gamma(y). \tag{114}$$

For Burg's entropy and $y = \nabla h^*(x) = -\frac{1}{x}$ with $x < 0$, this writes

$$\nabla^2 \eta_\gamma(x) = y^4 \nabla^2 \psi_\gamma(y) + 2y^3 \text{Diag}(\nabla \psi_\gamma(y)). \tag{115}$$

Using
$$\psi_\gamma(y) = -h(y) - g_\gamma(y) - 1, \tag{116}$$
$$\nabla\psi_\gamma(y) = -\nabla h(y) - \nabla g_\gamma(y) = \frac{1}{y} - \nabla g_\gamma(y) \tag{117}$$

and
$$\nabla^2\psi_\gamma(y) = -\nabla^2 h(y) - \nabla^2 g_\gamma(y) = -\frac{1}{y^2} - \nabla^2 g_\gamma(y), \tag{118}$$

we get
$$\begin{aligned}
\nabla^2\eta_\gamma(x) &= y^2\left(y^2\nabla^2\psi_\gamma(y) + 2yDiag(\nabla\psi_\gamma(y))\right) \\
&= y^2\left(-1 - y^2\nabla^2 g_\gamma(y) + 2 - 2y\nabla g_\gamma(y)\right) \\
&= y^2\left(1 - y^2\nabla^2 g_\gamma(y) - 2y\nabla g_\gamma(y)\right)
\end{aligned} \tag{119}$$

For $\eta_\gamma$ to be convex, a necessary condition is that $\forall y \in \mathbb{R}^n_{++}$ and $d \in \mathbb{R}^n$
$$\langle \nabla^2\eta_\gamma(y)d, d\rangle \geq 0 \tag{120}$$

i.e.
$$\langle y^4\nabla^2 g_\gamma(y)d, d\rangle \leq \sum_{i=1}^{n}\left(y^2(1 - 2y\nabla g_\gamma(y))\right)_i d_i^2 \tag{121}$$

**Training details**   For $N_\gamma$, we use the DRUNet architecture from [Zhang et al., 2021] with 2 residual blocks at each scale and softplus activation functions. We condition the network $N_\gamma$ on $\gamma$ similarly to what is done for DRUNet. We stack to 3 color channels of the input image an additional channel containing an image with constant pixel value equal to $1/\gamma$. We use the same training dataset as [Zhang et al., 2021]. Training is performed with ADAM during 1200 epochs. The learning rate is initialized with learning rate $10^{-4}$ and is divided by 2 at epochs 300, 600 and 900.

**Validation of the convexity hypothesis**   Given the trained Bregman Score Denoiser $\mathcal{B}_\gamma = y - y^2\nabla g_\gamma(y)$, we propose to verify validity of the hypothesis of convexity from Proposition 1 i.e. the convexity of $\eta_\gamma : x \to \psi_\gamma \circ \nabla h^*(x) = \psi_\gamma\left(\frac{-1}{x}\right)$ on $int\,\mathrm{dom}(h^*) = \mathbb{R}^n_{--}$. $\psi_\gamma$ is defined in (11). We showed above that for $z \in \mathbb{R}^n_{--}$ and $y = -1/z$, $\forall d \in \mathbb{R}^n$,
$$\langle \nabla^2\eta_\gamma(z)d, d\rangle = \sum_{i=1}^{n}\left(y^2(1 - 2y\nabla g_\gamma(y))\right)_i d_i^2 - \langle y^4\nabla^2 g_\gamma(y)d, d\rangle \tag{122}$$

In order to verify this assumption around the image manifold, we represent in Figure 4
$$c(\gamma, \xi) = \min_{x\in\{X_i\},d\in\{D_i\}}\left\langle \nabla^2\eta_\gamma\left(\frac{-1}{y_\xi}\right)d, d\right\rangle \tag{123}$$

$$= \min_{x\in\{X_i\},d\in\{D_i\}}\sum_{i=1}^{n}\left(y_\xi^2(1 - 2y_\xi\nabla g_\gamma(y_\xi))\right)_i d_i^2 - \langle y_\xi^4\nabla^2 g_\gamma(y_\xi)d, d\rangle \tag{124}$$

where the minimum is taken over

- $\{X_i\}_i$: the images from an evaluation dataset of clean images, here the 68 images from CSBD68.
- $\{D_i\}$: 100 random images realizations of a uniform random variable $D_i \sim \mathcal{U}[0,1]^n$.

and $y_\xi$ is obtained from $x$ by interpolating between $\hat{y}_\xi \sim p_\xi(y|x)$ a noisy version of $x$ and $\mathcal{B}_\gamma(\hat{y}_\xi)$ with $\alpha \sim \mathcal{U}[0,1]$ as
$$y_\xi = \alpha\hat{y}_\xi + (1-\alpha)\mathcal{B}_\gamma(\hat{y}_\xi). \tag{125}$$
In the figures, we represent $c(\gamma, \xi)$ with respect to:

- in the $x$-axis: the denoiser strength, i.e. the $\gamma$ parameter of the denoiser.
- in the $y$-axis: the noise level $\xi$ in the input image. Figure 4(a), the noise model is Inverse Gamma noise $p_\xi(y|x) = \mathcal{IG}(\xi - 1, \xi x)$ i.e. the noise model used for training the denoiser. In figure 4(b), the noise model is Poisson $p_\xi(y|x) = \mathcal{P}(\xi x)$ i.e. the noise model of the degradation on which our plug-and-play algorithms are evaluated, and in Figure 4(c), the noise model is Gaussian $p_\xi(y|x) = \mathcal{N}(x, \xi^2\,\mathrm{Id})$.

We observe that for a large variety of noise levels $\gamma$ and even far away from the image manifold i.e. for large noise, we verify the convexity assumption.

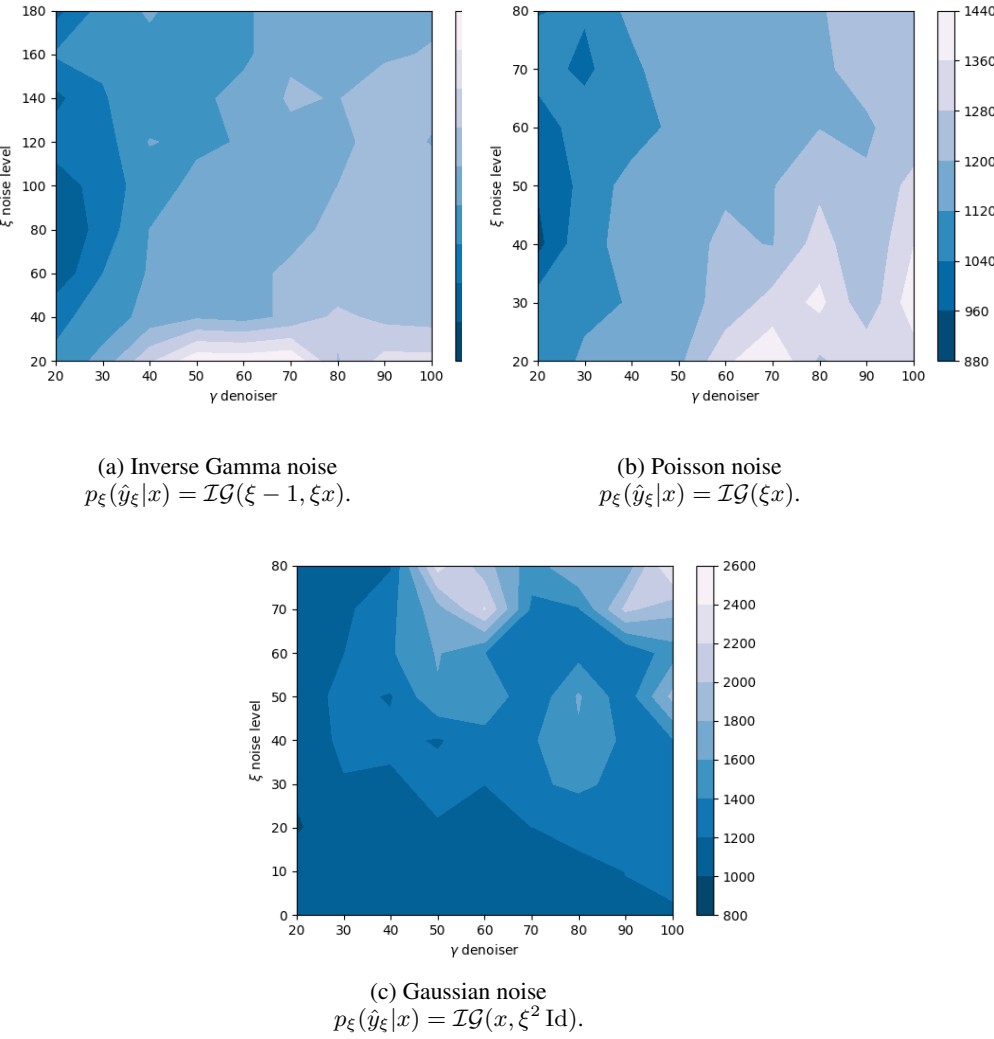

(a) Inverse Gamma noise
$p_\xi(\hat{y}_\xi|x) = \mathcal{IG}(\xi - 1, \xi x).$

(b) Poisson noise
$p_\xi(\hat{y}_\xi|x) = \mathcal{IG}(\xi x).$

(c) Gaussian noise
$p_\xi(\hat{y}_\xi|x) = \mathcal{IG}(x, \xi^2 \, \mathrm{Id}).$

Figure 4: Plot of $c(\gamma, \xi) = \min_{y_\xi, d} \left\langle \nabla^2 \eta_\gamma \left( \frac{-1}{y_\xi} \right) d, d \right\rangle$ w.r.t the denoiser parameter $\gamma$ and the noise level in the input image $\xi$. From $\{x_i\}$ the natural images from the CBSD68 testset and 100 random $d \sim \mathcal{U}[0,1]^n$, the image $y_\xi$ is obtained from $x \in \{x_i\}$ via the interpolation $y = \alpha \hat{y}_\xi + (1 - \alpha)\mathcal{B}_\gamma(\hat{y}_\xi)$ where $\hat{y}_\xi$ is a noisy version of $x$ sampled with (a) Inverse Gamma (b) Poisson (c) Gaussian noise distributions. $\mathcal{B}_\gamma$ is the trained Bregman Score Denoiser.

### E.3 The MMSE denoiser verifies the convexity hypothesis from Proposition 1

**Lemma 4.** *For $h$ Burg's entropy, and thus for the noise model* (23)*, with the MMSE denoiser $\hat{x}_{MMSE}(y) = E[x|y]$, we have $\eta_\gamma : x \to \psi_\gamma \circ \nabla h^*$ is strictly convex on $\mathrm{int} \, \mathrm{dom}(h^*)$, where $\psi_\gamma$ is defined in* (11) *from $g_\gamma = -\frac{1}{\gamma} \log p_Y$.*

*Proof.* Recall that the MMSE denoiser corresponds to the Bregman Score Denoiser

$$\mathcal{B}_\gamma(y) = \nabla(\psi_\gamma \circ \nabla h^*) \circ \nabla h(y) = \nabla \eta_\gamma \circ \nabla h(y) \tag{126}$$

in the particular case where we have exactly

$$g_\gamma(y) = -\frac{1}{\gamma} \log p_Y(y). \tag{127}$$

Note that in the Euclidean case (i.e. for the $L^2$ Bregman divergence $h = \frac{1}{2}||.||^2$ and $p_Y = p_\sigma = p * \mathcal{N}(0, \sigma^2 \mathrm{Id})$), it is shown in Gribonval [2011] to be verified by the MMSE denoiser. We now show the same result for Burg's entropy Bregman potential. For $x \in \mathbb{R}^n_{++}$

$$h(x) = -\sum_{i=1}^n \log(x_i), \tag{128}$$

Recall that, in this case, the Bregman noise model writes

$$p(y|x) = \exp(\rho(x) + n\gamma) \prod_{i=1}^n \left(\frac{x_i}{y_i}\right)^\gamma \exp\left(-\gamma\frac{x_i}{y_i}\right) = \alpha(x) \prod_{i=1}^n (y_i)^{-\gamma} \exp\left(-\gamma\frac{x_i}{y_i}\right). \tag{129}$$

where $\alpha(x) = \exp(\rho(x) + n\gamma) \prod_{i=1}^n x_i^\gamma > 0$. We verify that $\forall x \in int\, \mathrm{dom}(h^*) = \mathbb{R}^n_{--}$

$$\langle \nabla^2 \eta_\gamma(x)d, d \rangle > 0. \tag{130}$$

We showed in (119) that, $\forall y \in \mathbb{R}^n_{++}$

$$\nabla^2 \eta_\gamma(\nabla h(y)) = Diag\left(y^2(1 - 2y\nabla g_\gamma(y))\right) - y^4 \nabla^2 g_\gamma(y). \tag{131}$$

Similar to Gribonval [2011], for simplicity, we first write the proof in the single variable case ($n = 1$).

We have $\nabla g_\gamma(y) = \frac{-1}{\gamma}\frac{p'_Y(y)}{p_Y(y)}$ and $\nabla^2 g_\gamma(y) = \frac{1}{\gamma}\frac{[p'_Y(y)]^2 - p''_Y(y)p_y(y)}{p_Y^2(y)}$, and thus

$$\eta''_\gamma(x) = y^2\left(1 + \frac{2y}{\gamma}\frac{p'_Y(y)}{p_Y(y)} - \frac{y^2}{\gamma}\frac{[p'_Y(y)]^2 - p''_Y(y)p_y(y)}{p_Y^2(y)}\right) \tag{132}$$

$$= \frac{y^2}{\gamma p_Y^2(y)}\left(\gamma p_Y^2(y) + 2y p'_Y(y)p_Y(y) - y^2[p'_Y(y)]^2 + y^2 p''_Y(y)p_y(y)\right) \tag{133}$$

Moreover

$$p_Y(y) = \int_z \alpha(z)p_X(z)y^{-\gamma}\exp\left(-\gamma\frac{z}{y}\right)dz = \int_x u(z, y)dz. \tag{134}$$

$$p'_Y(y) = \frac{-\gamma}{y}p_Y(y) + \frac{\gamma}{y^2}\int_z zu(z, y)dz = \frac{-\gamma}{y}p_Y(y) + \frac{\gamma}{y^2}I^{(1)}(y) \tag{135}$$

where we denote $I^{(1)}(y) = \int_z zu(z, y)dz$.

$$p''_Y(y) = (\gamma^2 + \gamma)\frac{1}{y^2}p_Y(y) - 2(\gamma^2 + \gamma)\frac{1}{y^3}I^{(1)}(y) + \gamma^2\frac{1}{y^4}I^{(2)}(y), \tag{136}$$

where we denote $I^{(2)}(y) = \int_z z^2 u(z, y)dz$.

Eventually, we get, after simplification,

$$\eta''_\gamma(x) = \frac{y^2}{\gamma p_Y^2(y)}\frac{\gamma^2}{y^2}\left(I^{(2)}(y)p_Y(y) - [I^{(1)}(y)]^2\right) \tag{137}$$

$$= \frac{\gamma}{p_Y^2(y)}\int_z\int_{z'}\left(\frac{z^2}{2} + \frac{z'^2}{2} - zz'\right)u(z, y)u(z', y)dzdz' \tag{138}$$

$$= \frac{\gamma}{p_Y^2(y)}\int_z\int_{z'}\frac{(z - z')^2}{2}u(z, y)u(z', y)dzdz' \tag{139}$$

$$= \frac{\gamma y^{-2\gamma}}{p_Y^2(y)}\int_z\int_{z'}\frac{(z - z')^2}{2}\exp\left(-\gamma\frac{z + z'}{y}\right)\alpha(z)\alpha(z')p_X(z)p_X(z')dzdz' \geq 0 \tag{140}$$

As $\alpha(x) > 0$, the previous term is 0 if and only if $p_X(z)p_X(z') = 0$ when $z \neq z'$, which would imply $p_X(z) = 0\ \forall z$. This is impossible since $p_X$ is a proper pdf.

The extension for $n > 1$ is straightforward and $(z - z')^2$ becomes $\langle z - z', d \rangle^2$. $\qquad\square$

### E.4 Convergence of B-RED and B-PnP algorithm for Poisson inverse problems

In this section we verify that the Burg's entropy $h$ in (22) and the Poisson data likelihood $f$ defined in (2) verify the assumptions required for convergence of B-RED and B-PnP. We remind the expression of $f$ and $h$.

$$h(x) = -\sum_{i=1}^{n} \log(x_i), \tag{141}$$

$$f(x) = \sum_{i=1}^{m} y_i \log \left( \frac{y_i}{\alpha(Ax)_i} \right) + \alpha(Ax)_i - y_i, \tag{142}$$

for $A \in \mathbb{R}^{m \times n}$. Note that as done in Bauschke et al. [2017], denoting $(a_i)_{1 \leq i \leq n}$ the columns of $A$, we assume that $a_i \neq 0_m$ and $\forall 1 \leq j \leq m, \sum_{i=1}^{n} a_{i,j} > 0$ such that $Ax \in \mathbb{R}_{++}^m$ if $x \in \mathbb{R}_{++}^n$. This is verified for $A$ representing the blur with circular boundary conditions with the (normalized) kernels used in Section 5.2.

We first check **Assumptions 1**.

- (ii) $f$ is real analytic and thus subanalytic on $\mathbb{R}_{++}^n$.

- (iii) It is shown in [Bauschke et al., 2017, Lemma 7] that $f$ verifies the NoLip assumption, *i.e.* $L_f h - f$ is convex on $\mathbb{R}_{++}^n$, for $L_f \geq ||y||_1$. $y$ stands for the Poisson degraded observation appearing in the definition of $f$.

- (iv) First, $h$ is strongly convex everywhere on its domain except in $+\infty$. For $C$ a bounded subset of $\mathbb{R}_{++}^n$, as $\nabla^2 h(x) = \frac{1}{x^2} \mathrm{Id}$, we have $\forall x \in C, \forall d \in \mathbb{R}^n, \langle \nabla^2 h(x)d, d \rangle > \frac{1}{\sup_{x \in C} x^2} ||d||^2$ indicating that $h$ is strongly convex on bounded subsets of $\mathbb{R}_{++}^n$. Second, $h$ and $f$ are Lipschitz continuous everywhere on $\mathbb{R}_{++}^n$ except close to 0. For both B-RED and B-PnP $\Psi(x) \to +\infty$ when $x \to 0$ and $\{\Psi(x) \leq \alpha\}$ avoids the case $x \to 0$.

- (v) We remind the parameterizations $\mathcal{B}_\gamma = \mathrm{Id} - \nabla g_\gamma$ and $g_\gamma(y) = \frac{1}{2} ||x - N_\gamma(x)||^2$ with a U-Net $N_\gamma$ (with softplus activations). The softplus activation function being real analytic, using the fact that the composition and sum of real analytic functions are real analytic [Krantz and Parks, 2002], $N_\gamma$ and then subsequently $g_\gamma$ are real analytic functions.

  For the convergence of B-PnP, we also need to prove that $\phi_\gamma$ is real analytic (and thus subanalytic) on its domain. For this assumption to be verified, we require in Theorem 2, $\psi_\gamma \circ \nabla h^*$ not only to be convex, but to be *strictly convex*. Indeed, using the expression (64) $\mathcal{B}_\gamma(y) = \nabla(\psi_\gamma \circ \nabla h^*) \circ \nabla h(y)$, we have for $y \in \mathrm{int}\,\mathrm{dom}(h)$

$$J_{\mathcal{B}_\gamma}(y) = \nabla\left(\nabla(\psi_\gamma \circ \nabla h^*) \circ \nabla h(y)\right) \tag{143}$$

$$= \nabla^2 h(y).\nabla^2(\psi_\gamma \circ \nabla h^*)(\nabla h(y)). \tag{144}$$

  By bijectivity of $\nabla h^*$ between $\mathrm{int}\,\mathrm{dom}(h^*)$ and $\mathrm{int}\,\mathrm{dom}(h)$, we have

$$\psi_\gamma \circ \nabla h^* \text{ strictly convex on } \mathrm{int}\,\mathrm{dom}(h^*) \tag{145}$$

$$\Leftrightarrow \forall z \in \mathrm{int}\,\mathrm{dom}(h^*), \forall u \in \mathbb{R}^n, \ \langle \nabla^2(\psi_\gamma \circ \nabla h^*)(z)u, u \rangle > 0 \tag{146}$$

$$\Leftrightarrow \forall y \in \mathrm{int}\,\mathrm{dom}(h), \forall u \in \mathbb{R}^n, \ \langle \nabla^2(\psi_\gamma \circ \nabla h^*)(\nabla h(y))u, u \rangle > 0. \tag{147}$$

  Now, using the fact that $\nabla^2 h(y)$ is positive definite, it follows

$$\Leftrightarrow \forall y \in \mathrm{int}\,\mathrm{dom}(h), \forall u \in \mathbb{R}^n, \ \langle \nabla^2 h(y).\nabla^2(\psi_\gamma \circ \nabla h^*)(\nabla h(y))u, u \rangle > 0 \tag{148}$$

$$\Leftrightarrow \forall y \in \mathrm{int}\,\mathrm{dom}(h), \forall u \in \mathbb{R}^n, \ \langle J_{\mathcal{B}_\gamma}(y)u, u \rangle > 0 \tag{149}$$

$$\Leftrightarrow J_{\mathcal{B}_\gamma} \text{ positive definite on } \mathrm{int}\,\mathrm{dom}(h). \tag{150}$$

  By the inverse function theorem [Krantz and Parks, 2002], as $\forall y \in \mathrm{int}\,\mathrm{dom}(h), J_{\mathcal{B}_\gamma}(y) > 0$, $\mathcal{B}_\gamma^{-1}$ is then real analytic on $Im(\mathcal{B}_\gamma)$. We finally obtain, using the expression (12) of $\phi_\gamma$ on $Im(\mathcal{B}_\gamma)$, again by sum and composition, that $\phi_\gamma$ is real analytic (and thus subanalytic) on its domain.

  Eventually, $g_\gamma$ is non-negative and we now prove that we have $\forall y \in \mathbb{R}^n, \phi_\gamma(y) \geq g_\gamma(y)$. If $y \notin \mathrm{int}\,\mathrm{dom}(h)$, as $Im(\mathcal{B}_\gamma) \subset \mathrm{int}\,\mathrm{dom}(h), \phi_\gamma(y) = +\infty$, this is verified. If $y \in \mathrm{dom}(h)$,

as $D_h(y,y) = 0$, we have

$$\begin{aligned}
\phi_\gamma(y) &= \phi_\gamma(y) + D_h(y,y) \\
&\geq \phi_\gamma(\mathcal{B}_\gamma(y)) + D_h(\mathcal{B}_\gamma(y),y) \\
&= g_\gamma(y),
\end{aligned} \tag{151}$$

where the inequality comes from (13) and the last equality from (12). Therefore $\phi_\gamma$ is also lower-bounded.

Second, we verify **Assumption 2** required for the convergence of B-RED.

(i) $[0,R]^n$ is a non-empty closed, bounded, convex and semi-algebraic subset of $\mathbb{R}^n_{++}$.

(ii) With the parametrization $g_\gamma(y) = \frac{1}{2}||x - N_\gamma(x)||^2$ with a neural network $N_\gamma$. $g_\gamma$ can be shown to have Lipschitz gradient (see Hurault et al. [2021, Appendix B] for a proof). We can not show a global NoLip property for $g_\gamma$. However, as $\nabla g_\gamma$ is $Lip(g_\gamma)$-Lipschitz, we have $\forall x \in (0,R]^n, \forall d \in \mathbb{R}^n$,

$$\langle \nabla^2 g_\gamma(x)d, d \rangle \leq Lip(g_\gamma)||d||^2 \leq Lip(g_\gamma)R^2 \sum_{i=1}^n \frac{d_i^2}{x_i^2} = Lip(g_\gamma)R^2\langle \nabla^2 h(x)d, d \rangle, \tag{152}$$

which proves that, for $L_\gamma = Lip(g_\gamma)R^2$, $L_\gamma h - g_\gamma$ is convex on $(0,R]^n$.

**B-PnP additional assumptions** We finally discuss the additional assumptions required for the convergence of B-PnP in Theorem 2.

- $\mathrm{Im}(\mathcal{B}_\gamma) \subseteq \mathrm{dom}(h) = \mathbb{R}^n_{++}$. We train the denoiser $\mathcal{B}_\gamma$ to restore images in $[\epsilon, 1]^n$ (with $\epsilon = 10^{-3}$), the denoiser is thus softly enforced to have its image in this range. In practice, we empirically verify during the iterations that we always get $x_k > 0$.

- $\psi_\gamma \circ \nabla h^*$ *convex on* $int\, \mathrm{dom}(h^*)$. This is discussed in detail in Appendix E.2.

- $\mathrm{Im}(\nabla h - \lambda \nabla f) \subseteq \mathrm{dom}(\nabla h^*)$. We now show that this condition is true if $\lambda||y||_1 < 1$. For $x > 0$

$$\nabla h(x) - \lambda \nabla f(x) = -\frac{1}{x} - \lambda \nabla f(x) = -\frac{x + x\lambda \nabla f(x)}{x}. \tag{153}$$

Thus we need to verify that $\forall 1 \leq i \leq n, 1 + \lambda x_i \nabla f(x)_i > 0$. For $f$ Poisson data-fidelity term, using $(Ax)_j = \sum_{k=1}^n a_{j,k}x_k$, we have $\forall 1 \leq i \leq n$,

$$\nabla f(x)_i = \sum_{j=1}^m -y_j \frac{a_{j,i}}{\sum_{k=1}^n a_{j,k}x_k} + \alpha a_{j,i} \tag{154}$$

and

$$1 + \lambda x_i \nabla f(x)_i = 1 + \alpha\lambda \sum_{j=1}^m a_{j,i}x_i - \lambda \sum_{j=1}^m y_j \frac{a_{j,i}x_i}{\sum_{k=1}^n, a_{j,k}x_k}. \tag{155}$$

We assumed that $A$ has positive entries and $\sum_{j=1}^m a_{j,i} = r_i > 0$. Therefore, using $0 \leq \frac{a_{j,i}x_i}{\sum_{k=1}^n, a_{j,k}x_k} < 1$, we get

$$1 + \lambda x_i \nabla f(x)_i \geq 1 - \lambda||y||_1 \tag{156}$$

which is positive if $\lambda||y||_1 < 1$.

- *The stepsize condition* $\lambda L_f < 1$. Using the NoLip constant proposed in [Bauschke et al., 2017, Lemma 7] $L_f = ||y||_1$, the condition boils down to $\lambda||y||_1 < 1$. The condition $\lambda||y||_1 < 1$ is too restrictive in practice, as $||y||_1$ could get very big, especially for large images. This is due to the fact that the NoLip constant $L_f \geq ||y||_1$ can be largely over-estimated. Indeed, in the proof of [Bauschke et al., 2017, Lemma 7] as well as in the proof of the previous point, the upper bound

$$\frac{a_{j,i}x_i}{\sum_{k=1}^n, a_{j,k}x_k} < 1 \tag{157}$$

can be very loose in practice. For B-RED this is not a problem, as we use automatic stepsize backtracking. However, for B-PnP, backtracking is not possible as the stepsize is fixed to $\tau = 1$. In order to still guarantee the convergence, we adopt the following backtracking-like strategy to adjust the regularization parameter $\lambda$ :

- Choose an initial value for $\lambda > 0$.
- At each iteration of the B-PnP algorithm, check sufficient decrease of the objective function $F = \lambda f + \phi_\gamma$ i.e.

$$F(x_k) - F(x_{k+1}) \leq \delta D_h(x_{k+1}, x_k). \tag{158}$$

If at some iteration, this condition is not satisfied before convergence, we alert the user and restart the algorithm with $\lambda \longleftarrow \eta \lambda$ for some $\eta \in (0,1)$. We also let the user know that, for optimal performance, it might be necessary to adjust the regularization parameter $\gamma$ of the denoiser, in order to compensate for this decrease of $\lambda$.

In our deblurring experiments, for the proposed value of $\lambda$, over the variety of blur kernels and noise levels experimented, the sufficient decrease property was always verified and this backtracking algorithm was never activated. This illustrates that $||y||_1$ is a rough approximation of the NoLip constant.

### E.5 Experiments

We give here more details and results on the evaluation of B-PnP and B-RED algorithms for Poisson image deblurring. We present in Figure 5 the four blur kernels used for evaluation. Initialization is done with $x_0 = A^T y$. The algorithm terminates when the relative difference between consecutive values of the objective function is less than $10^{-8}$ or the number of iterations exceeds $K = 500$.

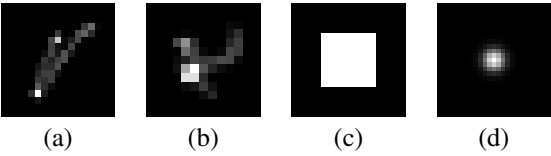

|        (a)        |        (b)        |        (c)        |        (d)        |

Figure 5: The 4 blur kernels used for deblurring evaluation. (a) and (b) are real-world camera shake kernels from Levin et al. [2009]. (c) is a $9 \times 9$ uniform kernel. (d) is a $25 \times 25$ Gaussian kernel with standard deviation 1.6.

**Choice of hyperparameters** For B-RED, stepsize backtracking is performed with $\gamma = 0.8$ and $\eta = 0.5$.

When performing plug-and-play image deblurring with our Bregman Score Denoiser trained with Inverse Gamma noise, for the right choice of hyperparameters $\lambda$ and $\gamma$, we may observe the following behavior. The algorithm first converges towards a meaningful solution. After hundreds of iterations, it can converge towards a different stationary point that does not correspond to a visually good reconstruction. We illustrate this behavior Figure 6 where we plot the evolution of the PSNR and of the function values $f(x_k)$ and $g_\gamma(x_k)$ along the algorithm.

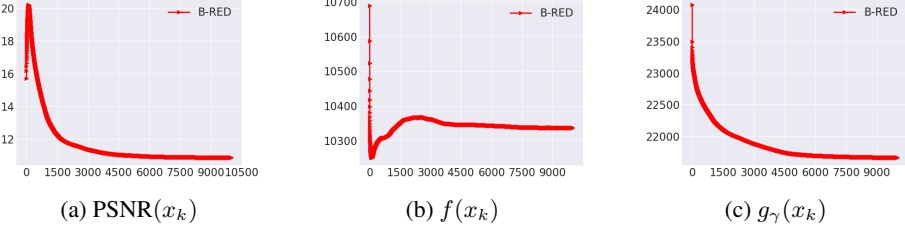

|        (a) $\mathrm{PSNR}(x_k)$        |        (b) $f(x_k)$        |        (c) $g_\gamma(x_k)$        |

Figure 6: Evolution of the PSNR, $f(x_k)$ and $g_\gamma(x_k)$ when deblurring with B-RED with the initialization parameters from Table 2 and without hyper-parameter update after 100 iterations. We observe a first phase of fast decrease of both the data-fidelity term and regularization term values, resulting in a fast PSNR increase. After approximately 100 iterations, the regularization continue decreasing and the iterates converge towards a different stationary point with low PSNR.

This phenomenon can be mitigated by using small stepsize, large $\gamma$ and small $\lambda$ values at the expense of slowing down significantly the algorithm. To circumvent this issue, we propose to first initialize the algorithm with 100 steps with initial $\tau$, $\gamma$ and $\lambda$ values and then to change this parameters for

the actual algorithm. Note that it is possible for B-RED to change the stepsize $\tau$ but not for B-PnP which has fixed stepsize $\tau1 =$. For B-PnP, as done in Hurault et al. [2022] in the Euclidean setting, we propose to multiply $g_\theta$ by a parameter $0 < \alpha < 1$ such that the Bregman Score denoiser becomes $\mathcal{B}^\alpha_\gamma(y) = y - \alpha(\nabla^2 h(y))^{-1} . \nabla g_\gamma(y)$. The convergence of B-PnP with this denoiser follows identically. The overall hyperparameters $\lambda, \gamma, \tau$ and $\alpha$ for B-PnP and B-RED algorithms for initialization and for the actual algorithm are given in Table 2.

| | $\alpha$ | 20 | 40 | 60 |
|---|---|---|---|---|
| B-RED | Initialization $\tau = 1 \ \gamma = 50$ | $\lambda = 1.5$ | $\lambda = 2.$ | $\lambda = 2.5$ |
| | Algorithm $\tau = 0.05 \ \gamma = 500$ | $\lambda = 0.5$ | $\lambda = 0.5$ | $\lambda = 0.5$ |
| B-PnP | Initialization $\alpha = 1 \ \gamma = 50$ | $\lambda = 1.5$ | $\lambda = 2.$ | $\lambda = 2.5$ |
| | Algorithm $\alpha = 0.05 \ \gamma = 500$ | $\lambda = 0.025$ | $\lambda = 0.025$ | $\lambda = 0.025$ |

Table 2: B-RED and B-PnP hyperparameters

**Additional experimental results** We provide in Table 3 a quantitative comparison between our 2 algorithms B-RED and B-PnP and 3 other methods. (a) PnP-PGD corresponds to the plug-and-play proximal gradient descent algorithm $x_{k+1} = D_\sigma \circ (\text{Id} - \tau \nabla f)$ with $D_\sigma$ the DRUNet denoiser (same architecture than B-RED and B-PnP) trained to denoiser Gaussian noise. (b) PnP-BPG corresponds to the B-PnP algorithm $x^{k+1} = D_\sigma \circ \nabla h^*(\nabla h - \tau \nabla f)(x_k)$ with again the DRUNet denoiser $D_\sigma$ trained for Gaussian noise. For both (a) and (b) the parameters $\sigma$ and $\tau$ are optimized for each noise level $\alpha$. (c) ALM Unfolded [Sanghvi et al., 2022] uses the Augmented Lagrangian Method for deriving a 3-operator splitting algorithm that is then trained specifically in an unfolded fashion for image deblurring with a variety of blurs and noise levels $\alpha$. The publicly available model being trained on grayscale images, for restoring our color images, we treat each color channel independently.

Note that contrary to the proposed B-PnP and B-RED algorithms, the 3 compared methods do not have any convergence guarantees. We observe that our algorithms performs the best when the Poisson noise is not too intense ($\alpha = 40$ and $\alpha = 60$) but that PSNR performance decreases for intense noise ($\alpha = 20$). We assume that this is due to the fact that the denoising prior trained on Inverse Gamma noise is not powerful enough for such a strong noise. A visual example for $\alpha = 20$ is given Figure 7. As a future direction, we plan on investigating how to increase the regularization capacity of the deep inverse gamma noise denoiser to better handle intense noise.

| $\alpha$ | 20 | 40 | 60 |
|---|---|---|---|
| PnP-PGD | 23.81 | 24.41 | 24.45 |
| PnP-BPG | **23.85** | 24.26 | 24.71 |
| ALM Unfolded [Sanghvi et al., 2022] | 23.39 | 23.91 | 24.22 |
| B-RED | 23.58 | **24.54** | **24.90** |
| B-PnP | 23.29 | **24.54** | 24.80 |

Table 3: PSNR (dB) of Poisson deblurring methods on the CBSD68 dataset. PSNR averaged over the 4 blur kernels represented Figure 5 for each noise levels $\alpha$.

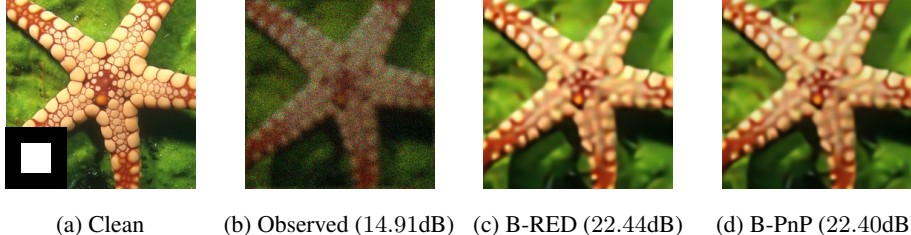

| (a) Clean | (b) Observed (14.91dB) | (c) B-RED (22.44dB) | (d) B-PnP (22.40dB) |
|---|---|---|---|

Figure 7: Deblurring from the indicated motion kernel and Poisson noise with $\alpha = 20$.

