# OpenReview forum: "Convergent Bregman Plug-and-Play Image Restoration for Poisson Inverse Problems"
_NeurIPS.cc/2023/Conference — NeurIPS 2023 poster_

### Official Review · Reviewer_oqFb · 2023-06-29

**Soundness:** 4 excellent
**Presentation:** 2 fair
**Contribution:** 3 good
**Rating:** 6
**Confidence:** 3

**Summary:**

Inspired by the No-Lips literature on the optimization of convex objectives which are not globally L-smooth, this paper adds to the PnP literature the “Bregman score denoiser” which extends the BPG algorithm by a Bregman-based prox-map along with convergence conditions despite NN-parametrized non-convex potentials are involved.

**Strengths:**

Originality:
This is a plausible extension of recent related work on PnP networks based on state-of-the-art theory from the field of numerical optimization.

Quality:
It is apparant that the authors do not both fields very well.

Significance:
The paper adds a new concept to the PnP literature.

**Weaknesses:**

Clarity:
The presentation intersperses references, top-level arguments and technical details in a confusing manner. I had to read few times forth and back in order to get an idea what this paper is about.

Authors criticize “unrealistic assumptions” (e.g. line 92) in related work but have to admit later on that their own assumptions are hard to check as well, even for a simple scenario (lines 275-277). The need for backtracking line search is not convenient either.

Significance:
Regarding the theory, I did not get if some `generic’ properties of the Bregman-prox-maps exist, that play the very same role like, say, the firmly-nonexpansiveness of Euclidean prox-maps, to achieve convergence in the considered generalized scenarios. The concrete scenario (Poisson noise) is classical. A comparison to a related approach shows no improvement (Fig.1, (c), (d)). In particular, artificially corrupting images looks like old-style image denoising papers.

**Questions:**

Before eq. (14), the convexity assumption regarding F and R, adopted in [Bauschke et al 2017] to which you refer, is missing.

line 92: why is nonexpansiveness considered as an “unrealistic requirement” to ensure convergence pf PnP, in view of your own assumptions which are not easy to check either?

How is the set C in (18) determined?

**Limitations:**

This point has not been explicitly addressed in the paper, apparently. Please comment.

---

> ### Author Rebuttal · Authors · 2023-08-05
>
> **Weaknesses:**
>
> - We first bring more details on our assumption of convexity of $\psi\_\gamma \circ \nabla h^\star(x)$, compared to the common assumption of nonexpansivity of the denoiser.
>
> It is observed in different works [1,2,3] that a state-of-the-art network trained to denoise without additional constraints is **not nonexpansive**, and constraining a network during training to be nonexpansive can severely degrade its denoising performance (see for example [1, Table 1], [2, Figure 1], [3, Table 5&6]). A semi-theoretical reason for this fact is that, by minimizing the $L^2$ cost, the denoiser is trained to approximate the true MMSE denoiser, which is *not nonexpansive*. This is why we call nonexpansivity an "unrealistic assumption”. Conversely, as proved in the global rebuttal to all reviewers, our assumption (convexity of $\psi\_\gamma \circ \nabla h^\star(x)$) **is verified by the true MMSE denoiser**, and is thus more realistic. This explains why, without any additional constraints, our trained denoiser verifies this assumption.
>
> We would like to put forward the distinction between *verifying (or checking)* the validity of the assumption, after training, and *enforcing* the assumption, while training. Note that our convexity condition is not *"hard to check"* but *hard to enforce explicitly*. Indeed, the convexity condition can be easily verified empirically, after training, with (26)-(27). Please refer to the global rebuttal to all reviewers, where we discuss and plot, in the attached pdf, empirical verifications of this convexity. We observe that the condition (26)-(27) is clearly verified even far way from the image manifold.
>
> In comparison, coming back to the nonexpansivity  assumption, finding the Lipschitz constant of a neural network being NP-hard, *verifying* the nonexpansivity of a neural network is typically done with a similar empirical verification. Moreover, previous methods *enforcing* nonexpansivity of a denoiser while training do not have better theoretical guarantees. Indeed, existing methods enforce nonexpansivity by either:
> - normalizing each layer by its spectral norm, with the spectral norm *approximated* via the power method [4]. As this is only a local approximation of the Lipschitz constant, there is no strict guarantee to get nonexpansivity.
> - regularizing the training loss [5,3] and thus without strict guarantee either.
>
> Let us also point out that the convexity condition is required for the convergence of the B-PnP algorithm but not for the convergence of B-RED. B-RED has then exact convergence guarantees.
>
> - Concerning the backtracking line search strategy, it allows to keep a fast algorithm along with convergence guarantees as it automatically sets the stepsize to its maximal value for convergence. Backtracking is commonly used when the Lipschitz (or NoLip) constant of the gradient of the smooth potential is unknown, which is typically the case when regularizing inverse problems with deep explicit priors, see for example [6].
>
> - Regarding the "generic" properties of the Bregman-prox-maps, Euclidean proximity operators of *nonconvex potentials* are not firmly-nonexpansive (this is actually why our denoiser is not nonexpansive) . The convergence of proximal algorithms like Proximal Gradient Descent in the nonconvex setting is only due to the *sufficient decrease property* obtained by combining the first order optimality of the prox and the descent lemma on the potential with Lipschitz gradient. This is the same idea for the proof for BPG algorithm with a Bregman prox, where the descent lemma holds thanks to the NoLip (4) property, which replaces the Lipschitz gradient assumption.
>
> - Concerning denoising, Figure 1 is not evaluating Poisson denoising but Inverse Gamma noise denoising. In this experiment, we do not expect the B-DRUNet denoiser to outperform the DRUNET denoiser. Indeed, both are based on the same architecture  but the former is additonally constrained to take the specific form (25). We are thus satisfied with the fact that this specific B-DRUNET almost reach the performance of DRUNET (difference of ~0.05 dB).
>
> - Eventually, regarding the artificial corruption of images, the purpose of training a denoiser using artificial Inverse Gamma noise (or artificial Gaussian noise for standard Plug-and-Play methods) is to establish a robust deep prior that effectively regularizes inverse problems. The selection of the (artificial) Bregman noise model holds significant importance in ensuring that our Bregman Score denoiser offers a meaningful prior. Once this prior is established through training, it becomes applicable in an unsupervised manner for any inverse problem, even for restoring images corrupted with real noise.
>
> **Questions:**
>
> - The NoLip condition will be added before eq (14), thanks for noticing this omission.
> - Concerning the "nonexpansivity considered as an unrealistic requirement”, please refer to our comment above.
> - In practice, we choose $C = [0,1]^n$.
>
> **Limitations:**
>
> Thanks for pointing this out. In the global rebuttal to all reviewers, we provide a limitation paragraph which will be added to the conclusion.
>
> [1] Hertrich et al. "Convolutional proximal neural networks and PnP algorithms". In Linear Algebra and its App., 2021.
>
> [2] Bohra et al. "Learning lipschitz-controlled activation functions in neural networks for PnP im. rec. methods". NeurIPS Workshop on DL and Inv Prob, 2021.
>
> [3] Hurault et al. "Proximal denoiser for convergent PnP optimization with nonconvex regul.". ICML 2022.
>
> [4] Ryu et al. "PnP methods provably converge with properly trained denoisers". ICML 2019.
>
> [5] Pesquet et al. "Learning maximally monotone operators for im. recovery." SIAM Journal on Im Sciences, 2021
>
> [6] Romano et al. "The little engine that could: Regularization by denoising (red)". SIAM Journal on Im Sciences, 2017.

---

> > ### Comment · Reviewer_oqFb · 2023-08-14
> >
> > Thanks for your response.
> > I am more concerned about theoretical aspects than about experimental performance. In this respect, your arguments are comprehensible. I stick to my positive score.

---

### Official Review · Reviewer_5jWQ · 2023-07-02

**Soundness:** 4 excellent
**Presentation:** 3 good
**Contribution:** 3 good
**Rating:** 6
**Confidence:** 4

**Summary:**

This paper studies an extension of the Plug-and-Play (PnP) framework for solving inverse imaging problems by considering descent schemes in metrics different from L2: Motivated by the fact that some data fidelity terms such as the Kullback-Leibler divergence allow for an efficient minimization with the Bregman Proximal Gradient (BPG) method (an extension of proximal gradient descent to arbitrary Bregman distances instead of squared L2), the authors develop a parametrization of a learnable denoiser which can be interpreted as a proximal operator (or descent step) of a cost function w.r.t. the underlying Bregman distance that combines well with a particular data fidelity term. Under some additional assumptions, this allows proving the convergence of the resulting Bregman PnP framework. Numerical experiments illustrate that the resulting scheme can successfully solve deblurring problems with Poisson noise.


**Strengths:**

The paper is technically sound and presents the technical construction of the Bregman PnP approach very well. It is an elegant solution that closes a gap missing in the PnP (and RED) framework with learnable priors. I found it very convincing in terms of its theory and it even derives some general (smaller) missing pieces for convergence beyond the seminal works by Bolte, Bauschke, Teboulle and co-authors.

Despite this strengths section being short in comparison to the weaknesses, I think that the strong theoretical contribution along with an illustration of the implementation outweighs the weaknesses in terms of (benchmark) results, such that I am leaning towards accepting the paper.

**Weaknesses:**

The numerical experiments/results are not very convincing from a practical point of view.
- First, the denoising performance of the dedicated network is not better than that of a plain network (might not be very important).
- Second, the main paper does not compare to the plain PnP approaches with L2 fidelity. The supplementary material reports tiny differences only (with the proposed approach being slightly worse for large noise and slightly better for small noise)
- According to line 275, the desired condition of $\phi_\gamma \circ \nabla h^*$ being convex is not enforced but seems to hold empirically when training the network. Thus, any convergence guarantee is lost.
- Figure 5 in the supplementary material shows a concerning risk of ending up with a bad result. The mitigation strategy of first running 100 iterations with a first set of parameters and then switching to a different set of parameters represents a significant amount of fine-tuning (possibly exceeding the number of hyperparameters and amount of fine-tuning used for the standard L2 PnP approach), such that even the small improvements in table 3 of the supplementary material need to be viewed with care.

Minor aspects:
- In 225 the author decide to do backtracking line search to avoid estimating the NoLip constant, but would backtracking on the (differentiable but not L-smooth) data fidelity term considered here not work in the L2 case?
- The authors mention that there is no result on the convergence of the PDHG method for nonconvex regularizers (line 81). As I was curious, I briefly searched online and found "Precompact convergence of the nonconvex Primal–Dual Hybrid Gradient algorithm" by Sun et al., Journal of Computational and Applied Mathematics, 2018. Precompactness of the primal variable (the image) would be easy to ensure if one restricts every value to [0,1]. Is their result applicable? (Honestly, I have not read the paper yet).
- The condition $\lambda L_f <1 $ in Theorem 2 seems to limit the amount of data fidelity one can use in order to still have a convergent algorithm - is this a limitation?


**Questions:**

Considering a difficult (2-stage) optimization with different parameters to avoid bad minimizers as shown in Fig. 5 of the supplement, a lack of strict convergence guarantee as the convexity condition cannot be enforced, and negligible difference to a plain PnP or RED approach in terms of the PSNR, what is the advantage of the proposed method?

**Limitations:**

I don't think there is a potential negative societal impact of this work. In terms of general limitations, I think the authors should be more open about the fact, that the use of the Bregman distance framework did not result in improved results over prior algorithmic schemes.

---

> ### Author Rebuttal · Authors · 2023-08-05
>
> **Weaknesses:**
> - We do no expect the B-DRUNet denoiser to outperform the DRUNET denoiser. Indeed, both are based on the same architecture but the former is additionally constrained to take the specific form (25). We are thus satisfied with the fact that B-DRUNET almost reach the performance of DRUNET (difference of ~0.05 dB).
> - We recognize that in terms of numerical performance, our algorithms compare with existing methods. Yet, our method stands alone in offering guarantees of convergence. We believe that the combination of comparable performance to these methods along with our convergence guarantees holds great promise. In fact, we have introduced the first convergent and effective technique for Poisson image restoration using a nonconvex and deep prior.
> - Indeed, the convexity condition (26)-(27) is not hard-coded in the network architecture. However, it is naturally verified in practice by the trained denoiser. Please refer to the global rebuttal to all reviewers, where we discuss and plot, in the attached pdf, empirical verifications of the convexity hypothesis. We note a clear confirmation of the condition (26)-(27), even when evaluated far from the image manifold. We also prove in the rebuttal to all reviewers that the assumption is **verified by the true MMSE denoiser**.  By minimizing the $L^2$ cost, the denoiser is actually trained to approximate this MMSE denoiser. This explains why our denoiser naturally satisfies the convexity condition without necessitating supplementary constraint. In comparaison, as explained in the introduction, most of the studies in the literature on the convergence of PnP methods assume *nonexpansive denoisers*. However, the **true MMSE denoiser is not nonexpansive**. Consequently, it is observed in different works [1,2,3] that a network trained to denoise without additional constraints is *not nonexpansive*. Furthermore, ensuring nonexpansivity of a denoiser while training is often achieved through the use of soft penalties or approximations (which also lack explicit guarantees) and significantly degrades denoising performance.
> - On the numerical side, we agree that the adopted two-step process is a limitation of our approach. The learned regularizer being nonconvex, the algorithms can be sensitive to initialization and for strong degradations the algorithm might not converge towards to right critical point if not initialized properly.
> - For Poisson noise, we can show that the data-fidelity term does not verify any NoLip condition (4) w.r.t the L2 metric. However, with Proposition 4 (Appendix D), the backtracking strategy is provably guaranteed to find a stepsize such that the objective function decreases, **provided** that a NoLip is satisfied for some $L>0$. Indeed, Proposition 4 is based on the sufficient decrease property (81) derived from the NoLip condition. Thus, in the L2 case, adding backtracking does not lead to a converging scheme.
> - We are indeed aware of the result from Sun et al. Even though very interesting, the precompactness is however not easy to enforce for the PDHG algorithm. Indeed, hard constraining the values of one variable in [0,1] amounts to add a proximal step (a projection) in the algorithm. In our B-RED algorithm, when adding the hard constraint $i_C$ (18) to the Bregman Gradient Descent algorithm (17), we still fit the general BPG algorithm (14). However, adding a proximal step in the PDHG algorithm does not correspond to some known algorithm.
> Note that one could prove that it is possible to ensure precompactness and thus convergence of nonconvex PDHG if we add the assumption that $f$ has Lipschitz gradient. Such a convergence result would indeed makes sense when compared to the recent convergence results of ADMM/DRS in the nonconvex setting that require one function to have Lipschitz gradient [4]. However, here for Poisson deblurring, $\nabla f$ is not Lipschitz and this is not applicable.
> - The constraint on $\lambda$  is indeed a limitation. Thanks for pointing this out. It is due to the fact that the denoiser writes as a proximal step *with stepsize $1$*. We are thus forced to keep a fixed stepsize $\tau=1$. The BPG condition for convergence $\tau \lambda L_f < 1$ then becomes $\lambda L_f < 1$ i.e. a constraint on $\lambda$. As a future work, we plan to explore solutions for relaxing this constraint, for instance by using a different algorithm than BPG with a loosen stepsize constraint.
>
> **Questions:**
>
> Compared to other Poisson image restoration methods, the main advantage of our algorithms is their convergence. First and foremost, the convexity condition is required for the convergence of the B-PnP algorithm but not for the convergence of B-RED. B-RED has then strict convergence guarantees. In addition, for the convergence of B-PnP, we refer to our comment above on the practical and semi-theoretical verification of the convexity condition.
>
> Furthermore, although our experiments focus on Poisson Inverse Problems, the core contribution of our work lies in its theoretical advancements. We present the first Bregman extension within the plug-and-play and Regularization-by-Denoising frameworks, supported by a strong theoretical foundation. This encompasses novel convergence findings for the Bregman Proximal Gradient algorithm (see Appendix D.1) and a new characterization of Bregman proximity operators (see Appendix C). These comprehensive theoretical outcomes extend beyond our immediate plug-and-play objectives, holding potential for diverse applications.
>
> [1] Hertrich et al. "Convolutional proximal neural networks and PnP algorithms". In Linear Algebra and its App., 2021.
>
> [2] Pesquet et al. "Learning maximally monotone operators for image recovery" SIAM J. on Im. Sc., 2022.
>
> [3] Hurault et al. "Proximal denoiser for convergent PnP optimization with nonconvex regularization". ICML 2022.
>
> [4] Themelis & Patrinos. "DRS and ADMM for nonconvex optimization: Tight convergence results." SIAM J. on Optim, 2020.

---

> > ### Comment · Reviewer_5jWQ · 2023-08-16
> > **Please make limitations clear in the revised version - otherwise: thanks, nice work!**
> >
> > Dear authors,
> >
> > thanks a lot for the detailed answers and the new section on the limitations! I think the restriction on $\lambda$ could be mentioned explicitly as well. Just to make sure: The constraint on $\lambda$ was respected in the experiments, right? Otherwise, this crucially needs to be pointed out.

---

> > > ### Author Response · Authors · 2023-08-17
> > >
> > > Thanks for the advise. We will update the limitation paragraph to mention the restriction on $\lambda$ for the B-PnP algorithm.
> > >
> > > The constraint $\lambda L_f<1$, which is specific to the B-PnP algorithm (B-RED converges without such constraint) , may not be respected in our experiments.
> > >
> > > The best estimation of a global NoLip constant $L_f$ for the Poisson data-fidelity term we could get is $L_f=||y||_1$ (see Appendix E.3). However, for an image, the value $||y||_1$ is large and the restriction on $\lambda < \frac{1}{||y||_1}$  leads to extremely small $\lambda$ values.
> > >
> > > This approximation of a *global* NoLip constant $L_f$ can be *locally* very lose. In particular, the majoration (127) used for estimating this constant, is, for most images, way over-estimated.
> > >
> > > In order to still guarantee the convergence, as mentioned Appendix E.3, we adopt the following backtracking-like strategy to adjust the reguralization parameter $\lambda$:
> > > - Choose an initial value for $\lambda > 0$
> > > - At each iteration $k$ of the B-PnP algorithm. Check sufficient decrease of the objective function $F_{\lambda,\gamma} = \lambda f + \phi_\gamma$ i.e.
> > >
> > > $F_{\lambda,\gamma}(x_{k}) - F_{\lambda,\gamma}(x_{k+1}) < \delta D_h(x_{k+1},x_k)$
> > >
> > > If at some iteration, this condition is not satisified before convergence, we alert the user and restart the algorithm with $\lambda \longleftarrow \eta \lambda$. We also let the user know that, for optimal performance, it might be necessary to adjust the regularization parameter $\gamma$ of the denoiser, in order to compensate for this decrease of $\lambda$
> > >
> > > With the proposed default value of $\lambda=0.025$, over the variety of blur kernels and noise levels experimented, the sufficient decrease property was always verified and this backtracking algorithm was never activated. This illustrates that $||y||_1$ is a bad approximation of the NoLip constant.
> > >
> > >
> > > In order to clarify this point we propose to include the previous discussion in a new Appendix and add the following paragraph in the main paper :
> > >
> > >
> > > In our experiments, the constraint $\lambda L_f<1$ of  the B-PnP algorithm may not be respected. The *global* NoLip constant $L_f$ can indeed be *locally* very lose. As explained in the Appendix, we can adopt a backtracking-like strategy on the regularization parameter $\lambda$ to ensure convergence. Nevertheless, with the proposed default value $\lambda=0.025$,  this backtracking algorithm was never  activated over the variety of blur kernels and noise levels experimented."

---

### Official Review · Reviewer_6Scw · 2023-07-06

**Soundness:** 3 good
**Presentation:** 3 good
**Contribution:** 3 good
**Rating:** 6
**Confidence:** 4

**Summary:**

This paper develops a Bregman Plug and Play image restoration algorithm for solving under-determined inverse problems in the presence of Poisson measurement noise. The framework trains the image denoising algorithms used within PnP iterations (the "Bregman Score Denoiser") to remove noise with an exponential distribution that depends on the distribution of the measurement noise (as opposed to the Gaussian noise used for training previous PnP denoisers). The paper integrates the proposed denoiser into iterative algorithms to form B-PnP and B-RED. Both algorithms effectively deblur images in the presence of Poisson noise and are provably convergent to fixed points.


**Strengths:**

-Generally well-written

-Well motivated

-I believe technically sound

-Evaluated on several different blur kernels, including those based on camera shake. (Blur kernels were known)


**Weaknesses:**

-Based on table 3 in the appendix, the proposed method does not meaningfully improve performance over existing methods (though it does come with convergence guarantees)

-Comparisons with existing methods are not particularly comprehensive

-In practice, PnP algorithms are quite sensitive to how hyperparameters are chosen and setting them correctly can be a challenge. This algorithm introduces another parameter, gamma.

**Questions:**

Typos
Line 90: "see also a review Kamilov et al." --> "see also a review by Kamilov et al."
Line 197: Puting (i) on a separate line would read cleaner


**Limitations:**

Not discussed.

---

> ### Author Rebuttal · Authors · 2023-08-05
>
> - We recognize that in terms of numerical performance, our algorithms compare with existing methods. Yet, as you correctly pointed out, our method stands alone in offering guarantees of convergence. We believe that the combination of comparable performance to these methods along with our convergence guarantees is a relevant novelty. In fact, we have introduced the first convergent and effective technique for Poisson image restoration using a nonconvex and deep prior. Furthermore, although our experiments focus on Poisson Inverse Problems, the core contribution of our work lies in its theoretical advancements. Our theoretical study contains novel convergence findings for the Bregman Proximal Gradient algorithm (see Appendix D.1) and a new characterization of Bregman proximity operators (see Appendix C). These comprehensive theoretical outcomes extend beyond our immediate plug-and-play objectives, holding potential for diverse applications.
>
> - Thanks for your feedback on the comparison section. In the updated version of the publication, we propose to bring more details when decribing the compared methods, as follows:
>
> a) PnP-PGD corresponds to the standard plug-and-play proximal gradient descent algorithm in the Euclidean geometry
> $$ x_{k+1} = D_\sigma \circ (Id - \tau \nabla f)(x_k) $$
> where $f$ is the Poisson data-fidelity term (2). The plugged denoiser $D_\sigma$ is the DRUNet  network (i.e. the same architecture than the one used for parametrizing our denoiser) but now trained to denoise **Gaussian** noise of std $\sigma$. We train $D_\sigma$  simultaneously for all noise levels $\sigma \in [0,50]$ with the $L^2$ loss. As $f$ does not have Lipschitz gradient, this PnP-PGD algorithm does not have convergence guarantees.
>
> b) PnP-BPG realizes the same iterations than our B-PnP algorithm (20)
> $$x^{k+1} = D_\sigma \circ \nabla h^*(\nabla h - \tau \nabla f)(x_k)$$
> but our Bregman Score Denoiser in (20) is replaced by the more classical Gaussian denoiser $D_\sigma$ introduced above. This scheme does not have guarantees of convergence as $D_\sigma$ is not a Bregman proximity operator anymore.
>
> For both PnP-PGD and  PnP-BPG the parameters $\sigma$ and $\tau$ are optimized for each noise level by grid-search.
>
> c) ALM Unfolded [3] uses the Augmented Lagrangian Method for decoupling linear operator $A$ from the data-fidelity term. They derive a 3-step algorithm that is unfolded and trained for specific degradations. In particular, it is trained for image deblurring with a variety of blurs and noise levels. The publicly available model being trained on grayscale images, for restoring our color images, we treat each color channel independently.
>
> - Concerning the hyperparameters involved in our algorithms, the parameter $\gamma$ is the noise level of the denoiser. $D_\gamma$ is trained to denoise images degraded with noise parameter $\gamma$. It is the equivalent of the parameter $\sigma$ commonly used by PnP method with Gaussian denoisers. Thus, compared to standard PnP algorithms, there is no additional hyperparameter involved. Actually, we even have one hyperparameter less as the stepsize of the algorithm has not to be chosen: for B-RED it is automatically tuned with a backtracking line-search strategy, and for B-PnP it has to be set to $1$. We agree on the fact that hyperparameters selection is a common difficulty in variational image restoration. Given an input image, automatic parameter tuning strategies have been developed and could be added to our framework. For example, [1] proposes a Bayesian method for setting the regularization parameter and [2] employs deep reinforcement learning to train a policy network yielding well-suited parameters for PnP.
>
> - In the global rebuttal to all reviewers, we propose a limitation paragraph which will be added at the end of the main paper.
>
> [1] Vidal et al. "Maximum likelihood estimation of regularization parameters in high-dimensional inverse problems: An empirical bayesian approach part i: Methodology and experiments.", 2020
>
> [2] Wei et al. "Tuning-free plug-and-play proximal algorithm for inverse imaging problems". In ICML 2020.
>
> [3] Sanghvi et al.  Photon limited non-blind deblurring using algorithm unrolling. IEEE Transactions on Computational Imaging, 2022.

---

> > ### Comment · Reviewer_6Scw · 2023-08-17
> >
> > Thanks for the response. The proposed comparisons with existing methods will improve the paper.

---

### Author Rebuttal · Authors · 2023-08-05

We would like to thank all reviewers for their careful reading of our submission, and their helpful comments and suggestions.  In each individual rebuttal, we tried to answer to all the objections raised by the reviewers. We give here more general remarks about aspects that multiple reviewers have highlighted.

**Regarding the hypothesis of convexity of $\psi\_\gamma \circ \nabla h^\star(x)$ in Proposition 1**

- We first proceed to a detailed experimental validation of this assumption (find the plots in the attached pdf):

After training the Bregman Score Denoiser $\mathcal{B}\_{\gamma}(y) = y - y^2 \nabla g\_\gamma(y)$ to denoise Inverse Gamma noise at different noise levels $\gamma$, we verify the convexity of $\eta_\gamma : x \to \psi_\gamma \circ \nabla h^\star(x) = \psi_\gamma\left(\frac{-1}{x}\right)$ on $int\ dom(h^\star) = \mathbb{R}^n\_{--}$ where $\psi_\gamma$ is defined in (11).

In appendix E.2, we showed that for $z \in \mathbb{R}^n\_{--}$ and $y = - 1/z$,  $\forall d \in \mathbb{R}^n$,

$\bigl \langle \nabla^2 \eta_{\gamma}(z) d,d \bigr \rangle = \sum_{i=1}^n \left( y^2(1 - 2y \nabla g\_\gamma(y)) \right)_i d_i ^2 - \bigl \langle Diag(y^4)\nabla^2 g\_\gamma(y)d, d \bigr \rangle = \mathcal{C}\_\gamma(y, d)$

To confirm the convexity assumption around the image manifold, we need to verify that the above quantity is positive for any image $y$. We represent, **in the attached pdf**

$c(\gamma, \xi) = \min_{y_\xi, d} \biggl \langle \nabla^2 \eta_{\gamma}\left(\frac{-1}{y_\xi}\right) d,d \biggr \rangle =  \min_{y_\xi, d} \mathcal{C}\_\gamma(y_\xi, d)$

The minimum is taken over $\\{y_{\xi}\\}$ and $\\{d\\}$ where
- Given $\\{ x_i \\}$ the CSBD68 testset of clean and natural images, $y_\xi$ is obtained from $x \in \\{x_i\\}$ by interpolating between $\hat y_\xi \sim p_\xi(y|x)$ a noisy version of $x$ and $\mathcal{B}\_{\gamma}(\hat y_\xi)$ via $y\_\xi = \alpha \hat y\_\xi + (1-\alpha) \mathcal{B}\_{\gamma}(\hat y\_\xi)$, where $\alpha \sim \mathcal{U}[0,1]$ . We use different noise models $p_\xi$ and noise levels $\xi$. This enables to explore the space around the image manifold.
- For each test image, $100$ random vectors $d$ are sampled from $\mathcal{U}[0,1]^n$.

In the figures, we represent $c(\gamma, \xi)$ w.r.t:
- $x$-axis: the $\gamma$ parameter of the denoiser.
- $y$-axis: the noise level $\xi$ in the input image. Figure (a), the noise model is Inverse Gamma noise $p_\xi(y|x) = \mathcal{IG}(\xi-1, \xi x)$ i.e. the noise model used for training the denoiser. Figure (b), the noise model is Poisson $p_\xi(y|x) = \mathcal{P}(\xi x)$, i.e. the noise model on which our plug-and-play algorithms are evaluated, and Figure (c), the noise model is Gaussian $p_\xi(y|x) = \mathcal{N}(x,\xi^2 Id)$.

We observe that for all  $\gamma$ seen during training, and even far away from the image manifold (i.e. for large noise level) we have $c(\gamma, \xi) > 0$  by a large margin, i.e. the convexity condition of Proposition 1 is clearly verified. These plots will be added to the appendix of the paper.

- We also provide a theoretical argument supporting the assumption:

Our denoiser is trained by minimizing the $L^2$ cost for which the optimal Bayes estimator is the MMSE. The MMSE is then a theoretical denoiser that our denoiser tries to approximate. **We can show that the proposed assumption ($\psi\_\gamma \circ \nabla h^\star(x)$ convex) is verified by the MMSE denoiser.** This explains why our denoiser, after training, naturally satisfies this condition without necessitating supplementary constraints. This result is proven in [1, Lemma A.1] in the Euclidean case, we here extend the proof for Burg's entropy Bregman potential.

We summarize the proof here without including the details of the calculus. We use the same notations as above. With $h(x) = \sum_{i=1}^n \log(x_i)$ Burg's entropy, it consists in showing that, for

$g_\gamma(y) = - \frac{1}{\gamma} \log p_Y(y)$ (i.e. when the denoiser is the MMSE)

we have

$\forall z \in \mathbb{R}^n_{--}, \langle \eta_\gamma(z) d,d \rangle \geq 0$.

We showed in Appendix E.2 that  $\forall z \in \mathbb{R}^n_{--}$ and  $y = \nabla h^*(x) = -\frac{1}{z}$,

$\nabla^2 \eta\_\gamma(z) = Diag \left(y^2(1 - 2y \nabla g\_\gamma(y)) \right) - Diag(y^4) \nabla^2 g\_\gamma(y)$

We consider the single variable case $n=1$. After differentiating $g_\gamma$ twice, we get

$\eta_\gamma''(z) = \frac{y^2}{\gamma p_Y^2(y)} \left( \gamma p_Y^2(y) + 2y p'_Y(y)p_Y(y) - y^2[p'_Y(y)]^2  + y^2 p''_Y(y)p_y(y) \right)$

Recall that for Burg's entropy, the Bregman noise model writes $p(y|x) = \alpha(x) \prod_{i=1}^n (y\_i)^{-\gamma} \exp\left(- \gamma \frac{x\_i}{y\_i}\right)$.
Differentiating $p_Y(y)= \int\_z p(y|z)p\_X(x)dx$ twice, after simplification we get

$\eta_\gamma''(z) = \frac{\gamma}{p_Y^2(y)} \int_x \int_{x'} \frac{(x-x')^2}{2} p(y|x)p(y|x')p(x)p(x')dxdx' \geq 0$

The extension to $n>1$ is straightforward, where $(x-x')^2$ becomes $\langle x-x', d \rangle^2$.

The detailed proof will be added to the appendix of the paper.

**Regarding the limitations, this section will be added in the main paper**:

The central significance of our work stems from its theoretical study but we recognize certain limits within our experimental results. First, when applied to deblurring with Poisson noise, our proposed algorithms do not outperform existing methods in terms of PSNR. Second, while we prove that B-RED is convergent without restriction, the convergence of B-PnP depends on a specific convexity condition. Despite being confirmed with experiments and having robust theoretical foundations, this assumption could potentially be not verified when applied to non-natural images that significantly differ from those in the training dataset. Finally, due to the nonconvex nature of our proposed prior, the practical performance of the algorithms can be sensitive to their initialization."

[1] Gribonval, "Should penalized ...." IEEE Trans on Signal Proc., 2011.

---

### Decision · Program_Chairs · 2023-09-21

**Decision:**

Accept (poster)

**Comment:**

The paper develops plug-n-play methods for inverse problems whose losses are not gradient-Lipschitz. The most significant example of this is inverse problems with Poisson noise, which are prevalent in medical and scientific imaging. The paper develops a framework for this problem, in which one trains a ``Bregman score denoiser’’, which at reconstruction time can be deployed as a component in a Bregman proximal method, with provable convergence.

Reviewers found the paper’s proposals to be well motivated, addressing a need in the PnP/RED frameworks. Reviewers found the paper to be theoretically solid, well written, and sufficiently novel to merit acceptance. Minor concerns involved the experimental results (where the proposed methods match, but do not exceed, the state of the art), and the existence of prior art on learnable inverse problem solvers with poisson noise. In this regard, the main contributions of the paper are (i) to develop a framework with provable convergence, and (ii) proving convergence. After considering the authors responses, the reviewers converged to a recommendation to accept the paper.